# Deep Learning and Foundation Models for Weather Prediction: A Survey

## Abstract

Conventional numerical weather prediction models are computationally expensive, while deep learning approaches often offer faster, and sometimes more accurate predictions. However, challenges do persist. This paper presents a comprehensive survey of recent deep learning and foundation models for weather prediction. We propose a taxonomy to classify existing models based on their training paradigms: *deterministic predictive* learning, *probabilistic generative* learning, and *pre-training and fine-tuning*. For each paradigm, we delve into the underlying model architectures, address major challenges, offer key insights. Furthermore, we explore related applications of these methods and provide a curated summary of open-source code repositories and widely used datasets, aiming to bridge research advancements with practical implementations. Finally, we propose potential future research directions in this fast-growing field. The related sources are anonymously available at https://anonymous.4open.science/r/Survey.

## 1 Introduction

Accurate and timely weather prediction is critical for mitigating the impacts of extreme weather events (Rummukainen, 2012)and supporting decision-making across sectors such as agriculture, transportation, and disaster management (Abbass et al., 2022). Physics-based models, including General Circulation Models (GCMs) (Ravindra et al., 2019) and Numerical Weather Prediction (NWP) models (Coiffier, 2011), have been the cornerstone of weather prediction. These models simulate future weather scenarios by numerically approximating solutions to the differential equations that govern the complex physical dynamics of interconnected atmospheric, terrestrial, and oceanic systems (Nguyen et al., 2023a).

Despite significant advancements, these physics-based models face notable challenges. Firstly, the accuracy of conventional NWP models is highly dependent on the spatial and temporal resolution. Finer resolutions allow for better representation of mesoscale and localized phenomena (e.g., convection, topographic effects, or coastal systems). However, achieving higher resolution significantly increases the computational cost (Al-Yahyai et al., 2010). Secondly, subgrid-scale parameterizations introduce significant uncertainty. Many atmospheric processes occur at scales too small to be explicitly resolved and are instead approximated using empirical or simplified physical models. These parameterizations are often region-specific and may fail to generalize across varying conditions (Palmer et al., 2005). Lastly, a single physics-based model typically produce deterministic forecasts once initial conditions are fixed, falling short of capturing uncertainties in weather evolution even though perturbation of initial conditions has been used (Bülte et al., 2024). Ensemble-based NWP forecasts help address this issue by generating probabilistic outputs (Gneiting & Raftery, 2005), the first two challenges persist.

In recent years, data-driven machine learning (ML) and deep learning (DL) models have been increasingly applied to weather and climate modeling, demonstrating remarkable advances in precision, computational efficiency, and uncertainty quantification (Chen et al., 2023d; Nguyen et al., 2023b). They have proven increasingly adept at capturing complex atmospheric dynamics in an end-to-end fashion, eliminating the reliance on explicit prior knowledge of physical relationships. For example, deterministic models such as `Pangu` (Bi et al., 2023) and `GraphCast` (Lam et al., 2022) have achieved state-of-the-art performance in

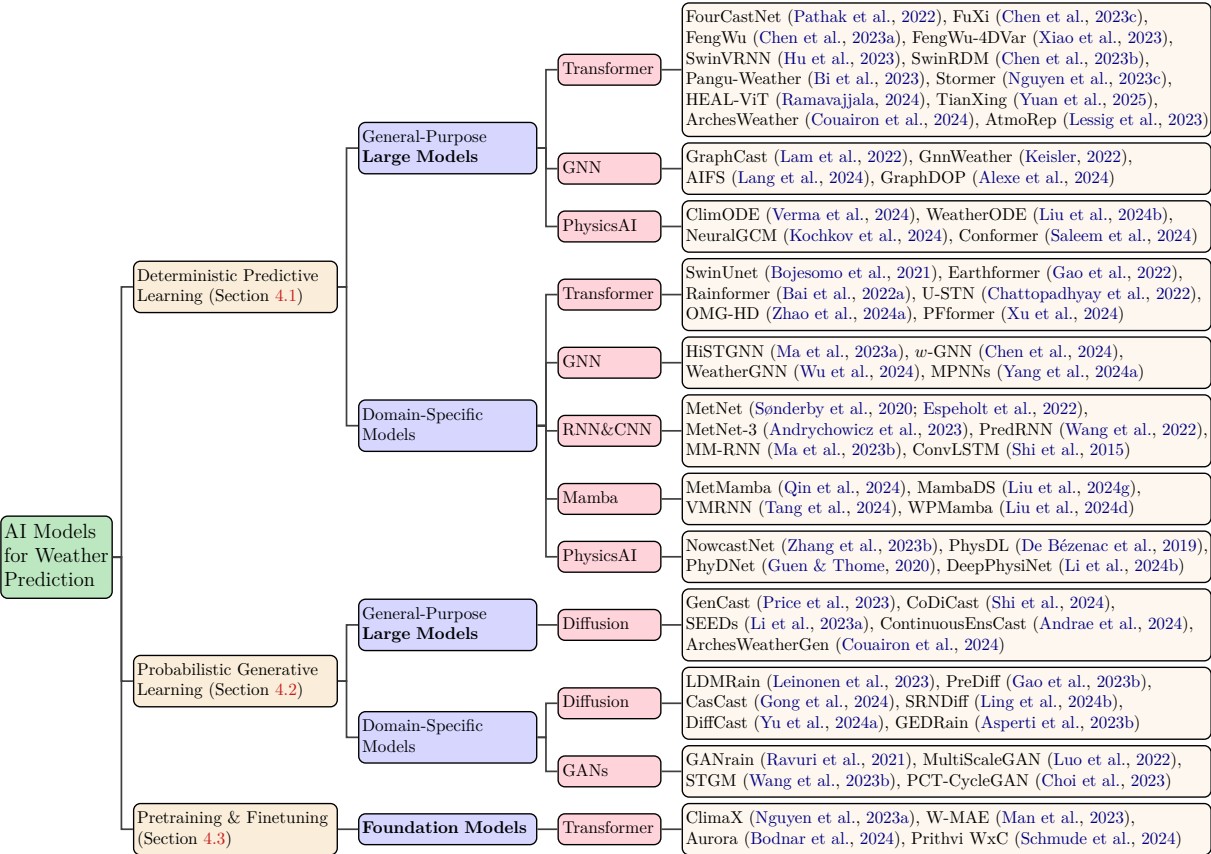

Figure 1: A comprehensive taxonomy of deep learning and foundation models for weather prediction from the perspectives of training paradigms (dark yellow), model scopes (purple), and model architectures (pink).

medium-range (10-day) global weather prediction, surpassing or matching traditional methods in terms of accuracy on some benchmark datasets (e.g., ERA5) while dramatically reducing computational costs (up to three orders of magnitude). However, their predictions are often blurry since they are trained by minimizing point-wise loss functions. To overcome this limitation, probabilistic generative models have emerged as powerful tools for weather prediction while achieving uncertainty quantification in those predictions. They consider weather prediction as probabilistic sampling (i.e., generation) conditioning on necessary constraints. Models like CasCast (Gong et al., 2024) and Gencast (Price et al., 2023) leverage diffusion models for precipitation nowcasting and weather prediction, delivering both probabilistic outputs and calibrated uncertainty estimates. More recently, foundation models have gained traction in climate and weather modeling as an emerging paradigm (Bodnar et al., 2024; Schmude et al., 2024). These models are pre-trained on massive historical weather datasets to learn generalizable and comprehensive knowledge, which can then be fine-tuned for diverse downstream tasks, e.g., weather forecasting and climate downscaling (Chen et al., 2023f). Foundation models offer two key advantages: (1) the ability to learn robust and transferable weather representations from large-scale data, and (2) the flexibility to adapt to downstream applications without the need for task-specific models trained from scratch (Miller et al., 2024; Zhu et al., 2024b).

With the rapid advancement of deep learning (DL) in weather and climate science, a systematic and up-to-date survey is essential for consolidating knowledge and guiding future research. Distinct from the existing surveys (Ren et al., 2021; Molina et al., 2023; Fang et al., 2021; Materia et al., 2024; Mukkavilli et al., 2023), our work provides a novel perspective by reviewing the literature through the lens of training paradigms. In summary, we propose a systematic taxonomy of existing DL models for weather prediction based on their training paradigms: *deterministic predictive learning, probabilistic generative learning*, and *pre-training and fine-tuning*. Building on this framework, we provide a comprehensive survey of state-of-the-art models, critically analyzing their strengths, limitations, and applicability to various forecasting tasks. To support con-

tinued progress in this domain, we compile a curated repository of resources, including benchmark datasets, open-source implementations, and real-world applications. Lastly, we outline a forward-looking roadmap, highlighting critical potential research opportunities for advancing the field of weather forecasting.

## 2 Related Surveys

Ren et al. (2021) reviewed DL models for weather prediction, with a focus on their architectural designs. Molina et al. (2023) explored DL applications in climate modeling, including feature detection, extreme weather prediction, downscaling, and bias correction. Other surveys, such as those by Fang et al. (2021) and Materia et al. (2024), concentrated on DL techniques for specific scenarios, such as forecasting extreme weather events. Additionally, Mukkavilli et al. (2023) highlighted state-of-the-art DL models across diverse meteorological applications, emphasizing their performance across various spatial and temporal scales. Moreover, Chen et al. (2023f) categorized DL models for weather and climate science based on data modalities (e.g., time series, text) and their respective applications.

## 3 Background

### 3.1 Weather Data Representation

Weather forecasting and climate modeling rely on a variety of data sources, each offering distinctions in terms of spatial and temporal resolution, coverage, and observational depth. The primary categories of weather data include *station-based observations*, *gridded reanalysis datasets*, and *remote sensing data from radar and satellite* platforms.

**Station-Based Observation Data.** Station-based observations are collected from a global network of meteorological stations, which record high-resolution measurements at specific geographic locations. These stations provide key variables such as temperature, humidity, wind speed and direction, precipitation, atmospheric pressure, and solar radiation. Due to their high temporal resolution (typically hourly or sub-daily), station data are crucial for analyzing short-term weather events and validating model forecasts. However, the spatial distribution of weather stations is highly uneven, with denser coverage in urban and economically developed regions and sparse representation in remote areas such as oceans, polar regions, and mountainous terrain. This spatial heterogeneity limits the ability to conduct comprehensive global analyses using station data alone.

**Gridded Reanalysis Data.** Gridded reanalysis datasets offer a coherent, global representation of past atmospheric states by assimilating multiple data sources, including station observations, satellite data, and outputs from numerical weather prediction (NWP) models. These datasets discretize the Earth's surface into uniform grids, typically with resolutions ranging from $1° \times 1°$ to $0.25° \times 0.25°$ (each degree corresponds to about 100 km), enabling large-scale spatial analysis. Reanalysis products such as ERA5 (Hersbach et al., 2020), MERRA-2 (Gelaro et al., 2017), and JRA-55 (Kobayashi et al., 2015) are widely used for climate diagnostics, model benchmarking, and long-term trend analysis. Temporal resolution can vary from hourly to daily, supporting multi-scale applications across both weather and climate domains.

**Radar and Satellite Remote Sensing Data.** Radar and satellite observations provide critical information on atmospheric processes over regions where ground-based data are limited or unavailable. Radar systems, primarily ground-based, are instrumental in monitoring high-frequency precipitation events, storm dynamics, and convective systems at fine spatial ($\sim$1 km) and temporal ($\sim$5–10 minutes) scales. These data are particularly valuable for short-term forecasting (nowcasting) and extreme weather monitoring, such as tracking thunderstorms or flash floods. Satellite platforms, such as those operated by NOAA, NASA, and EUMETSAT, offer broad, continuous coverage of atmospheric, oceanic, and land surface conditions. They capture a range of variables, including cloud properties, sea surface temperatures, outgoing longwave radiation, and atmospheric moisture profiles. Both geostationary and polar-orbiting satellites contribute to operational forecasts, with products available at varying spatial and temporal resolutions. The integra-

tion of radar and satellite data into machine learning models enhances predictive accuracy, particularly for precipitation forecasting, cyclone tracking, and global-scale anomaly detection.

## 3.2 Weather Prediction Tasks

As shown in Figure 2, we discuss weather forecasting tasks from the following four perspectives. (1) *Temporal*: forecasts predict atmospheric variables of interest for future time point(s), $t + \Delta t$, given observation(s) from the recent past. It includes weather and climate forecasts based on the lead time $\Delta t \approx$ {hours, days, weeks, months, years} and encompasses nowcast, medium-range forecast, sub-seasonal, and seasonal forecast. Nowcasting predicts weather in the next few hours, medium-range forecasting covers days to two weeks, and seasonal prediction focuses on climate patterns over months, each differing in lead time, scale, and data requirements. (2) *Spatial*: methods predict global and regional weather forecasts for any given time point. (3) *Applications*: focus on predicting weather variables of interest. (4) *Event Type*: Weather forecasts may be for extreme events, such as heatwaves, snowstorms, hurricanes, tropical cyclones, heavy rainfall, etc.

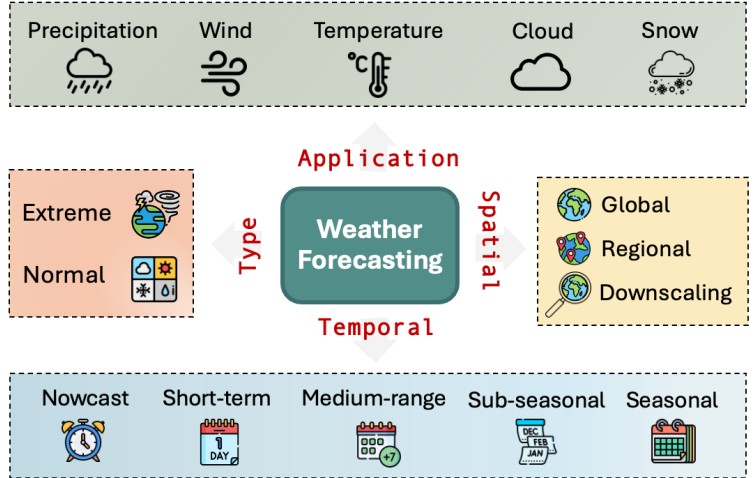

Figure 2: Perspectives of weather forecasting.

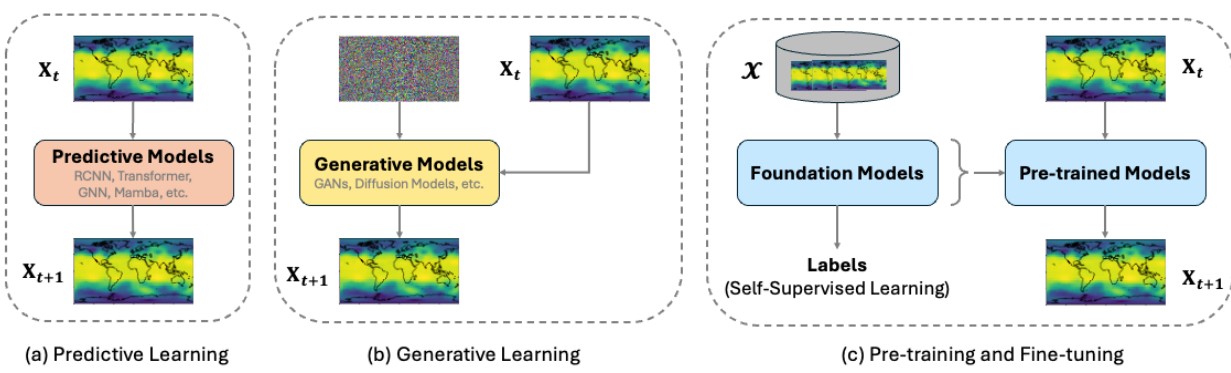

Figure 3: The illustration of various frameworks of training deep learning models on weather prediction. For clarity, this visualization focuses exclusively on single-step predictions for a single variable.

## 4 Overview and Taxonomy

This section presents an overview and systematic categorization of deep learning (DL) models for weather forecasting. Our survey is structured around three key dimensions: modeling paradigm, model architecture

(backbone), and application domains. Firstly, we distinguish models based on their modeling paradigms, which include: (1) deterministic predictive learning (Section 4.1), (2) probabilistic generative learning (Section 4.2), and (3) pre-training and fine-tuning strategies (Section 4.3). A high-level comparison is illustrated in Figure 3, and a detailed taxonomy is provided in Figure 1.

Secondly, these models can be categorized based on model backbones, such as Recurrent Neural Networks, Transformers, Graph Neural Networks, Mamba, Generative Adversarial Networks, and Diffusion Models. The theoretical details of these models are listed in Appendix B.

Thirdly, at the application level, existing DL weather forecasting models are divided into general-purpose and domain-specific models. General-purpose models are designed for global, multi-variable forecasting and typically operate at coarse spatial resolutions ($0.25°$–$5.625°$) and temporal intervals (6 or 12 hours). These models are trained on extensive historical datasets ($\geq 10$ years) and primarily employ Transformer and Graph Neural Network (GNN) architectures. Given the computational demands of training on global-scale data, even with GPU or TPU acceleration, coarser resolutions but longer periods remain a practical choice under current hardware constraints. This is also the common experimental setting for current global weather forecasting models in Figure 1. In contrast, domain-specific models focus on regional or single-variable forecasting tasks and operate at higher spatial ($\leq 0.1°$) and temporal (5 minutes $\sim$ 1 hour) resolutions. The smaller spatial coverage allows for the use of much finer-resolution data, enabling more detailed predictions. More importantly, it is helpful to capture localized and fine-grained spatiotemporal dynamics, which is the primary focus for those models targeting regional weather forecasting. A wider range of architectures is commonly adopted, including Transformers, GNNs, CNNs, RNNs, and Mamba-based models.

Table 1: General-Purpose Large Models vs Domain-Specific Models.

|  | General-Purpose Large Models | Domain-Specific Models |
|---|---|---|
| Scope | Global, multi-variable | Regional forecasts, single-variable |
| Spatial | Coarse ($0.25° \sim 5.625°$) | High ($\leq 0.1°$) |
| Temporal | Coarse ($6, 12$ hours) | High (5 mins $\sim$ 1 hour) |
| Training Data | $\geq 10$ Years | Days, Months, Years |
| Architectures | Transformer, GNN | Transformer, GNN, RNN, CNN, Mamba |

### 4.1 Predictive Learning

*Predictive* learning methods are usually *deterministic*, where models aim to predict future states of weather variables (like temperature, humidity, wind speed, and precipitation) based on past and present observations. These models are typically built to recognize weather patterns or dependencies in historical data by minimizing a point-wised loss function (e.g., mean absolute errors). We systematically categorize these predictive models into general-purpose large models and domain-specific models. Each categorization is discussed with various model architectures.

#### 4.1.1 General-Purpose Large Models

Large Language Models (LLMs) (Zhao et al., 2023) have garnered significant attention in recent years. Similarly, large-scale weather models have been developed to address global weather prediction tasks across multiple meteorological variables, leveraging deterministic predictive frameworks.

**Transformer-based models.** Transformer models (Vaswani, 2017) are widely used as a backbone. FourCastNet (Pathak et al., 2022) is developed for global data-driven weather forecasting by employing a Fourier transform-based token-mixing scheme (Guibas et al., 2021) with a vision transformer (ViT) (Dosovitskiy et al., 2020). The multiple-time step prediction is achieved by using trained models in autoregressive inference mode. FengWu (Chen et al., 2023a) introduces a multi-encoder design that processes each meteorological variable independently, followed by a transformer-based fusion network that captures inter-variable dependencies. This design aims to preserve variable-specific dynamics while learning complex inter-variable

relationships, which is particularly important for multivariate forecasting tasks. `FengWu-4DVar` (Xiao et al., 2023) extends `FengWu` with the Four-Dimensional Variational (4DVar) assimilation algorithm (Rabier et al., 1998), accomplishing both global weather forecasting and data assimilation. `SwinVRNN` (Hu et al., 2023) integrates the Swin Transformer (Liu et al., 2022) and RNN for weather prediction. The hierarchical structure of Swin Transformers is well-suited for capturing multi-scale spatial patterns, while RNNs are used to model temporal evolution. Additionally, a perturbation module is introduced to generate ensemble forecasts, offering a practical way to quantify uncertainty, a critical component for operational use. This makes it a notable contribution to the area of probabilistic weather forecasting. SwinRDM (Chen et al., 2023b) builds upon the `SwinVRNN` design but focuses on super-resolution forecasting. It uses the SwinRNN architecture for coarse-grained prediction and applies a diffusion model to upscale outputs to high-resolution forecasts. This two-stage design leverages the generative power of diffusion models for fine-grained detail synthesis, addressing the resolution gap that often limits operational usability in data-driven models. `HEAL-ViT` (Ramavajjala, 2024) explores Vision Transformers on a spherical mesh, benefiting from both spatial homogeneity inherent in graphical models and efficient attention mechanisms. The `TianXing` model (Yuan et al., 2025) proposes a variant attention mechanism with linear complexity for global weather prediction, significantly diminishing GPU resource demands, with only a marginal compromise in accuracy. The above models generate multi-step forecasts in an autoregressive manner, where the model is recursively applied at inference time.

Despite impressive performance, any iterative inference process accumulates errors as the length of the prediction window increases. To mitigate this phenomenon, the `Pangu-Weather` (Bi et al., 2023) model uses a hierarchical temporal aggregation algorithm to alleviate cumulative forecast errors. They train four individual models for lead times of 1, 3, 6, and 24 hours. In the testing stage, given a forecast goal with a certain lead time, the greedy algorithm is used to guarantee the minimal number of iterations of the trained models for that forecast window. For example, for a 7-day forecast, `Pangu` executes the 24-hour forecast 7 times, while for a 23-hour forecast, `Pangu` executes the 6-hour forecast 3 times, followed by the 3-hour forecast 1 time and the 1-hour forecast 2 times. Furthermore, they design a 3D Earth Specific Transformer (3DEST) architecture that formulates the height (pressure level) information into cubic data, capturing more intricate spatiotemporal dynamics. Similarly, the `FuXi` model (Chen et al., 2023c) employed a combination of FuXi-Short, FuXi-Medium, and FuXi-Long models to produce 15-day forecasts, where each model generates 5-day forecasts. Its backbone is a U-transformer, coupling U-Net (Ronneberger et al., 2015), and a Swin Transformer (Liu et al., 2022). In addition to the integration of direct and iterative forecasting, the `Stormer` model (Nguyen et al., 2023c) needs the explicit time point, $t + \Delta t$ to guide the models for predictions.

**GNN-based models.** Keisler (2022) introduced an approach to global weather prediction using graph neural networks (GNNs) (Wu et al., 2020). By modeling the Earth as a graph with nodes representing spatial locations and edges encoding their relationships, the model captures spatial dependencies in weather patterns. This GNN-based method effectively integrates local and global weather dynamics. Another GNN-based model, `GraphCast` (Lam et al., 2022), forecasts hundreds of weather variables with a longer forecast range (up to 10 days ahead) at a higher spatial resolution (0.25 degree) after training with reanalysis gridded ERA5 data (Rasp et al., 2023). It also provides better support for severe weather compared to the European Centre for Medium-Range Weather Forecasts (ECMWF)'s High-RESolution forecast (HRES), a component of the Integrated Forecast System (IFS). More recently, ECMWF also proposed GNN-based models, `AIFS` (Lang et al., 2024) and `GraphDOP` (Alexe et al., 2024). The latter is a model that operates solely on inputs and outputs in observation space, with no gridded climatology and/or NWP (re)analysis inputs or feedback. GNNs are particularly well-suited for weather forecasting due to their capacity to model irregular spatial structures, capture non-local dependencies, and generalize across heterogeneous spatial domains. However, they also present several limitations. First, designing graph topologies that faithfully represent atmospheric dynamics is nontrivial; fixed spatial graphs may fail to capture evolving spatiotemporal relationships and multi-scale interactions. Second, GNNs often face challenges in modeling long-range temporal dependencies, which are critical for medium- to long-range forecasts, unless explicitly augmented with temporal modules such as recurrent networks or attention mechanisms. Moreover, scaling GNNs to high-resolution global datasets can be computationally demanding due to the overhead of message passing, especially in densely connected or large-scale graph structures.

**Physics-AI-based models.** Although data-driven methods have demonstrated high accuracy and efficiency, they operate as black-box models that frequently overlook underlying physical mechanisms, such as turbulence, convection, and atmospheric airflow. `ClimODE` (Verma et al., 2024) implements a key principle of *advection* to model a spatiotemporal continuous-time process, namely, weather changes due to the spatial movement over time. It aims to precisely describe the value-conserving dynamics of weather evolution with continuity ODE (Marchuk, 2012), learning global weather transport as a neural flow. It also includes a Gaussian emission network for predicting uncertainties and source variations. To solve the advection equation more accurately, `WeatherODE` (Liu et al., 2024b) adopts wave equation theory (Evans, 2022) and a time-dependent source model and designs the CNN-ViT-CNN sandwich structure, facilitating efficient learning dynamics tailored for distinct yet interrelated tasks with varying optimization biases. `NeuralGCM` (Kochkov et al., 2024) employs a differentiable dynamical core for solving *more* primitive equations, including momentum equations, the second law of thermodynamics, a thermodynamic equation of state, continuity equation, and hydrostatic approximation. It also develops a learned physics module that parameterizes physical processes with a neural network, predicting the effect of unresolved processes such as cloud formation, radiative transport, precipitation, and subgrid-scale dynamics. `Conformer` (Saleem et al., 2024) is a spatiotemporal Continuous Vision Transformer for weather forecasting, learning the continuous weather evolution over time by implementing continuity in the multi-head attention mechanism. By explicitly incorporating governing equations or physical constraints, these hybrid models improve physical consistency, particularly in capturing conservation laws and flow dynamics. They are especially effective in regimes where data is sparse or noisy, leveraging physical knowledge to regularize the learning process.

### 4.1.2 Domain-Specific Models

We present domain-specific predictive models for regional or single-variable weather predictions.

**Transformer-based models.** `SwinUnet` (Bojesomo et al., 2021) employs the hybrid model of Swin Transformer and U-Net for regional weather forecasts in Europe. `Earthformer` (Gao et al., 2022) proposes a generic, flexible, and efficient space-time attention block (Cuboid Attention) Earth system forecasting, which can decompose the data into cuboids and apply cuboid-level self-attention in parallel. `Rainformer` (Bai et al., 2022a) combines CNN and Swin Transformer for precipitation nowcasting. `PFformer` (Xu et al., 2024) utilizes i-Transformer (Liu et al., 2023a) to learn spatial dependencies among multiple observation stations for short-term precipitation forecasting. Vision transformer (Dosovitskiy et al., 2020) has been applied to estimate lightning intensity in Ningbo City, China (Lu et al., 2022). `NowcastingGPT` (Meo et al., 2024) develops Transformer-based models with Extreme Value Loss (EVL) regularization (von Bortkiewicz, 1921) for extreme precipitation nowcasting. The `U-STN` model (Chattopadhyay et al., 2022) integrates data assimilation with a deep spatial-transformer-based U-NET to predict the global geopotential while the `OMG-HD` model (Zhao et al., 2024a) leverages the Swim Transformer for regional high-resolution weather forecast trained with multiple observational data, including stations, radar, and satellite.

**GNN-based models.** `HiSTGNN` (Ma et al., 2023a) incorporates an adaptive graph learning module comprising a global graph representing regions and a local graph capturing meteorological variables for each region. The $w$-`GNN` model (Chen et al., 2024) leverages Graph Neural Networks coupled with physical factors for precipitation forecast in China. `WeatherGNN` (Wu et al., 2024) proposes a fast hierarchical Graph Neural Network (FHGNN) to extract the spatial dependencies. The `MPNN` model (Yang et al., 2024a) exploits heterogeneous GNNs for both station-observed and gridded weather data, where the node at the prediction location aggregates information from its heterogeneous neighboring nodes by message passing.

**RNN- & CNN-based models.** The `ConvLSTM` model (Shi et al., 2015) couples CNNs and LSTMs as the model backbone for precipitation nowcasting, usually with a lead time between 1 to 3 hours. Similar works include `MetNet-1` (Sønderby et al., 2020) and `MetNet-2` models (Espeholt et al., 2022) for precipitation forecasting for lead times of 8 and 12 hours. `MetNet-3` (Andrychowicz et al., 2023) significantly extends both the lead times (up to 24 hours) and variables (precipitation, wind, temperature. `MM-RNN` (Ma et al., 2023b) introduces knowledge of elements to guide precipitation prediction and learn the underlying atmospheric motion laws using RNNs. Based on the original LSTMs, `PredRNN` (Wang et al., 2022) proposes a zigzag

memory flow that propagates in both a bottom-up and top-down fashion across all layers, enabling the dynamic communication at various levels of RNNs. Other variants of ConvLSTM for precipitation nowcasting include `TrajGRU` (Shi et al., 2017) and `Predrnn++` (Wang et al., 2018).

**Mamba-based models.** `MetMamba` (Qin et al., 2024) exploits Mamba's selective scan to achieve token (spatial, temporal) mixing and channel mixing to capture more complex spatiotemporal dependencies in weather data. `MambaDS` (Liu et al., 2024g) attempts to use the selective state space model (Mamba) for the meteorological field downscaling. `VMRNN` (Tang et al., 2024) develops an innovative architecture tailored for spatiotemporal forecasting by integrating Vision Mamba and LSTM, surpassing established vision models in both efficiency and accuracy. We observed that `Mamba` (Gu & Dao, 2023) has thus far been applied primarily to domain-specific tasks. Given its high efficiency in modeling long-range dependencies with linear computational complexity, it is promising for extension to global-scale weather forecasting.

**Physics-AI-based models.** `NowcastNet` (Zhang et al., 2023b) is a nonlinear nowcasting model for extreme precipitation that unifies physical-evolution schemes and conditional-learning methods into a neural network framework. `PhysicsAI` (Das et al., 2024) has evaluated `NowcastNet` model with a case study on the Tennessee Valley Authority (TVA) service area, outperforming the High Resolution Rapid Refresh (HRRR) model. `PhysDL` (De Bézenac et al., 2019) presents how physical knowledge (*advection* and *diffusion*) could be used as a guideline for designing efficient DL models, exemplifying sea surface temperature predictions. `PhyDNet` (Guen & Thome, 2020) is a two-branch deep learning architecture that explicitly disentangles known PDE dynamics from unknown complementary information. `DeepPhysiNet` (Li et al., 2024b) incorporates atmospheric physics into the loss function of deep learning methods as hard constraints for accurate weather modeling.

More generally, we provide state-of-the-art predictive models for time series forecasting across various domains. While these models are not specific for weather modeling, they offer insightful modeling advancements since weather data is often represented as time series. Representative models include but not limited to `iTransformer` (Liu et al., 2023a), `PatchTST` (Nie et al., 2022), `DLinear` (Zeng et al., 2023), `Autoformer` (Chen et al., 2021a). More recently, Han et al. (2024b) collected worldwide meteorological monitoring data, created a benchmark dataset, and completed a comprehensive evaluation with those advanced models above.

## 4.2 Generative Models

Generative models can be used for weather *prediction* by treating them as *generative* processes conditioned on observations from the past. More significantly, since these generative models are probabilistic, they are well suited to generate ensemble forecasts that can help quantify the uncertainty in the predictions, facilitating informed decision-making.

### 4.2.1 General-Purpose Large Models

**Diffusion-based models.** Some researchers have developed generative models for global weather prediction. `GenCast` (Price et al., 2023) uses diffusion models for probabilistic weather forecasts conditioning on the past two observations, generating an ensemble of stochastic 15-day global forecasts, at 12-hour steps and 0.25° latitude-longitude resolution, for over 80 surface and atmospheric variables. As a variant of `GenCast`, `CoDiCast` (Shi et al., 2024) leverages a *pre-trained* encoder to learn embeddings from observations from the recent past and a *cross-attention* mechanism to guide the generation process to predict future weather states. Similar work includes `SEEDs` (Li et al., 2023a) for the global weather forecast. The three methods above are trained on a single forecasting step and rolled out autoregressively. However, they are computationally expensive and accumulate errors for high temporal resolution due to the many rollout steps. `ContinuousEnsCast` (Andrae et al., 2024) addresses these limitations by proposing a continuous forecasting diffusion model that takes lead time as input and forecasts the future weather state in a single step while maintaining a temporally consistent trajectory for each ensemble member. `ArchesWeatherGen` (Couairon et al., 2024) first introduces a deterministic model for weather forecasting, and then enhances it with probabilistic forecasting capabilities by modeling the residuals—i.e., the differences between its predictions and ERA5 data—using flow matching, a variant of diffusion models, to generate ensemble forecasts.

#### 4.2.2 Domain-Specific Models

**GAN-based models.** GANrain (Ravuri et al., 2021) employs a conditional generative adversarial network (GAN) for the precipitation prediction problem, where the generator generates future precipitation frames and the discriminator learns to distinguish whether a sample is coming from the original training data or was generated by the generator. MultiScaleGAN (Luo et al., 2022) evaluates GANs (Goodfellow et al., 2014) and Wasserstein-GAN (Arjovsky et al., 2017) for precipitation nowcasting in Guangdong province, China, and indicates that GAN-based models outperform the traditional ConvGRU, ConvLSTM, and multiscale CNN models. STGM (Wang et al., 2023b) introduces a task-segmented, synthetic-data generative model (STGM) for heavy rainfall nowcasting by utilizing real-time radar observations in conjunction with physical parameters derived from the Weather Research and Forecasting (WRF) model. PCT-CycleGAN (Choi et al., 2023) extends the idea of the cycle-consistent adversarial networks (CycleGAN) (Zhu et al., 2017) and proposes a paired complementary temporal CycleGAN for radar-based precipitation nowcasting. Despite their widespread applications, GANs remain challenging to train due to instability and mode collapse issues. Moreover, we observe that GANs have been predominantly applied to domain-specific tasks, and their effectiveness for global-scale weather forecasting remains an open question.

**Diffusion-based models.** LDMRain (Leinonen et al., 2023) uses the architecture of latent diffusion model (Rombach et al., 2022) for precipitation nowcasting – short-term forecasting based on the latest observational data. Similar works include SRNDif (Ling et al., 2024b) and GEDRain (Asperti et al., 2023b). DiffCast (Yu et al., 2024a) models the precipitation process from two perspectives: the deterministic component accounts for predicting a global motion trend by a coarse forecast, while the stochastic component aims to learn local stochastic variations with the residual mechanism. CasCast (Gong et al., 2024) develops a cascaded framework consisting of a deterministic predictive model to output blurry predictions, and a probabilistic diffusion model with inputs as both past observations and deterministic predictions beforehand. Because the deterministic predictions are the future frames, such frame-wise guidance in the diffusion model can provide a frame-to-frame correspondence between blurry predictions and latent vectors, resulting in a better generation of small-scale patterns. However, directly applying diffusion models might generate physically implausible predictions. To tackle these limitations, Prediff (Gao et al., 2023b) proposes a conditional latent diffusion model for probabilistic forecasts and then aligns forecasts with domain-specific physical constraints. This is achieved by estimating the deviation from imposed constraints at each denoising step and adjusting the transition distribution accordingly.

TimeDiff (Shen & Kwok, 2023), TimeDDPM (Dai et al., 2023), LTD (Feng et al., 2024b), TimeGrad (Rasul et al., 2021), and Dyffusion (Rühling Cachay et al., 2024) are examples that have applied diffusion models to general time series modeling, which could be adapted to weather time series. Yang et al. (2024b) provides a comprehensive survey of such methods for time series and spatiotemporal modeling.

### 4.3 Foundation Models

Foundation Models (FMs) have garnered significant research interest due to their powerful prior knowledge acquired through pre-training on massive data and their remarkable adaptability to downstream tasks with fine-tuning strategies (He et al., 2024c). While foundation models may be large language models (LLMs), a few foundation models in the weather domain have been proposed.

ClimaX (Nguyen et al., 2023a) is a versatile and generalizable deep-learning model developed for weather and climate science. It is trained on heterogeneous datasets encompassing diverse variables, spatiotemporal coverage, and physical principles with CMIP6 datasets and it can be fine-tuned for a wide range of weather and climate applications, including those involving atmospheric variables and spatiotemporal scales not encountered during pre-training. W-MAE (Man et al., 2023) is pre-trained with similar data, but using reconstruction tasks with the Masked Autoencoder model (He et al., 2022). The pre-trained model can be fine-tuned for various tasks, e.g., multi-variate forecasting. Aurora (Bodnar et al., 2024) is a large-scale foundation model pre-trained on over a million hours of diverse weather and climate data. Unlike the two foundation models above, Aurora can be fine-tuned in one of two ways: short-time fine-tuning (i.e., fine-tuning the entire architecture through one or two roll-out steps) and rollout fine-tuning for long-term multi-

step predictions with low-rank adaption (LoRA) (Hu et al., 2021a). `Prithvi WxC` (Schmude et al., 2024) is a foundation model with 2.3 billion parameters developed using 160 variables. It is essentially a scalable and flexible 2D vision transformer with varying sizes of tokens or windows. During the pre-training, the Masked Autoencoder model (He et al., 2022) is pre-trained by masking different ratios of tokens and windows to capture both regional and global dependencies in the input data. It can be fine-tuned for nowcasting, forecasting, and downscaling tasks. More recently, `AtmosArena` (Nguyen et al., 2024) benchmarks foundation models for atmospheric sciences across various atmospheric variables.

Furthermore, time series foundation models designed for diverse domains may be flexibly adapted for weather forecasting. Representative examples include `TimeFM` (Das et al., 2023), `Moment` (Goswami et al., 2024), `Timer` (Liu et al., 2024e), `Moirai` (Woo et al., 2024), and `Chronos` (Ansari et al., 2024).

## 4.4 Quantitative Comparison and Discussion

In Table 2, we present a detailed quantitative comparison of a part of above models across three categories. More details can be found on the WeatherBench scorecard[1].

We break down our primary analysis as follows. First, probabilistic generative and foundation models are emerging as promising directions, although they remain in the early stages of development. Notably, `GenCast` has achieved state-of-the-art performance, surpassing many deterministic models while producing probabilistic forecasts. However, existing probabilistic models are primarily based on diffusion models, resulting in significantly longer inference times than deterministic alternatives. Second, foundation models are distinguished by pre-training on diverse and large-scale datasets (e.g., ERA5, CMIP6, HRTS-T0), setting them apart from previous methods that rely on a single dataset. This diversity enhances their generalizability and benefits the subsequent fine-tuning phase, allowing them to adapt effectively to new tasks and domains. Third, while ML-based models typically involve longer training durations and require substantial computational resources, they offer substantial speedups at inference time compared to traditional physics-based models such as IFS HRES. For instance, inference with models like `Pangu-Weather` or `GraphCast` takes seconds to minutes on modern GPUs or TPUs, in contrast to the 52 minutes required by IFS HRES. Fourthly, there exists a notable trade-off between spatial resolution ($\Delta x$) and performance metrics such as T850 and Z500. High-resolution models like `Pangu-Weather` and `GraphCast` (0.25°) generally exhibit superior performance on these metrics. However, certain lower-resolution models, including `GnmWeather` and `ArchesWeather`, achieve comparable Z500 scores. This suggests that advances in model architecture and the use of diverse training data can compensate for reduced spatial resolution to some extent.

Table 2: Comparison of Predictive, Generative, and Foundation Models for **global** weather prediction. The performance scores below are at the lead time of 6 days (except Prithvi WxC at the lead time of 5 days). These scores are either from the WeatherBench scoreboard or the original paper. "$\Delta x$" represents the horizontal resolution.

| Methods | $\Delta x$ | Train data | Train resources | Test data | Inference time | T850 $[K]$ | Z500 $[m^2/s^2]$ | U10m $[m/s]$ | V10m $[m/s]$ |
|---|---|---|---|---|---|---|---|---|---|
| *Physics-based Models* | | | | | | | | | |
| IFS HRES (ECMWF) | 0.1° | | | ERA5 2020 | ~52 mins | 2.23 | 411.07 | 3.05 | 3.17 |
| IFS ENS (ECMWF) | 0.2° | | | ERA5 2020 | – | 1.9 | 360.85 | 2.51 | 3.61 |
| *Deterministic Predictive Models* | | | | | | | | | |
| Pangu-Weather (Bi et al., 2023) | 0.25° | ERA5 1979-2017 | 16 days; 192 V100 GPUs | ERA5 2020 | ~secs; a GPU | 2.11 | 394.96 | 2.84 | 3.11 |
| GraphCast (Lam et al., 2022) | 0.25° | ERA5 1979-2019 | 4 weeks; 32 TPU v4 | ERA5 2020 | ~min; a TPU | 1.98 | 375.62 | 2.71 | 2.82 |
| FuXi (Chen et al., 2023c) | 0.25° | ERA5 1979-2015 | 8 days; 8 A100 GPUs | ERA5 2020 | ~secs; a GPU | 1.84 | 352.74 | 2.43 | 2.54 |
| Fengwu (Chen et al., 2023a) | 0.25° | ERA5 1979-2017 | 17 days; 32 A100 GPUs | ERA5 2020 | ~secs; a GPU | 1.97 | 365.62 | | – |
| Stormer (Nguyen et al., 2023c) | 0.25° | ERA5 1979-2017 | 8 days; 8 A100 GPUs | ERA5 2020 | ~secs; a GPU | 1.92 | 373.75 | 2.55 | 2.68 |
| HEAL-ViT (Ramavajjala, 2024) | 0.25° | ERA5 1979-2017 | 8 days; 8 A100 GPUs | ERA5 2020 | ~secs; a GPU | 1.99 | 369.99 | 2.72 | 2.98 |
| GnmWeather (Keisler, 2022) | 1° | ERA5 35 years | 5.5 days; 1 A100 GPU | ERA5 2020 | ~secs; a GPU | 2.14 | 403.84 | – | – |
| ArchesWeather(Couairon et al., 2024) | 1.5° | ERA5 1979-2018 | 9 days; 1 V100 GPU | ERA5 2020 | ~secs; a GPU | 1.98 | 381.05 | 2.66 | 2.79 |
| NeuralGCM 0.7 (Kochkov et al., 2024) | 0.7° | ERA5 1979-2017 | 3 weeks; 256 TPUs v5 | ERA5 2020 | ~min; a TPU | 1.98 | 363.88 | – | – |
| NeuralGCM ENS (Kochkov et al., 2024) | 1.4° | ERA5 1979-2017 | 10 days; 128 TPUs v5 | ERA5 2020 | ~min; a TPU | 1.82 | 345.86 | – | – |
| *Probabilistic Generative Models* | | | | | | | | | |
| GenCast (Price et al., 2023) | 0.25° | ERA5 1979-2018 | 5 days; 32 TPUs v5 | 13 | 8 mins; a TPU | 1.78 | 342.04 | 2.38 | 2.49 |
| ArchesWeatherGen(Couairon et al., 2024) | 1.5° | ERA5 1979-2018 | 45 days; 1 V100 GPU | 13 | – | 1.81 | 351.64 | 2.41 | 2.53 |
| *Foundation Models with Pre-training and Fine-tuning* | | | | | | | | | |
| Aurora (Bodnar et al., 2024) | 0.1°; 0.25° | ERA5, HRTS-T0, CMIP6, ... | 2.5 weeks; 32 A100 GPUs | HRES-T0 2022 | – | ~1.85 | ~350.22 | ~2.67 | – |
| Prithvi WxC(Schmude et al., 2024) | 0.5° | MERRA-2 1980-2019 | –; 64 A100 GPUs | MERRA-2 2020-2023 | – | ~2.25 | – | ~3.15 | – |

---

[1] https://sites.research.google/weatherbench/

# 5 Applications and Resources

This section introduces the diverse applications of deep learning models in weather and climate science. We provide an overview of the available datasets, summarized in detail in Table 3 in Appendix A.

## 5.1 Precipitation

Precipitation prediction has witnessed significant advances driven by deep learning (DL) applications, focusing mainly on precipitation nowcasting (Gao et al., 2020; 2021; Ashok & Pekkat, 2022; Verma et al., 2023; Salcedo-Sanz et al., 2024; An et al., 2024). CNN-based architectures, particularly U-Net, have been widely utilized for their ability to extract local features through convolutional layers, effectively capturing high-dimensional spatio-temporal dynamics of precipitation (Lebedev et al., 2019; Ayzel et al., 2020b; Han et al., 2021; Ehsani et al., 2022; Seo et al., 2022; Kim et al., 2022a; Zhang et al., 2023b). RNN-based models, Transformers, and their hybrid designs combining convolutions represent another dominant approach, optimized for long-term dependency modeling (Shi et al., 2015; Wang et al., 2017; Park et al., 2022; Gao et al., 2022; Bai et al., 2022a; Geng et al., 2024; Bodnar et al., 2024; Zhao et al., 2024b; Schmude et al., 2024). Generative models have also played a critical role, with adversarial models (e.g., GANs) (Jing et al., 2019; Liu & Lee, 2020; Ravuri et al., 2021; Wang et al., 2023c; She et al., 2023; Choi et al., 2023; Yin et al., 2024; Franch et al., 2024) contributing to precipitation synthesis. Moreover, probabilistic generative diffusion models have gained attention for their superior stability, controllability, and fine-grained synthesis capabilities (Leinonen et al., 2023; Gao et al., 2023b; Yu et al., 2024a; Gong et al., 2024).

## 5.2 Air Quality

Air quality prediction is of critical importance to society. Zheng et al. (2013) employ artificial neural network (ANN) with spatially-related features to predict the air quality in Beijing, Waseem et al. (2022) employed a CNN-Bi-LSTM architecture for air quality prediction in Xi'an, China, and Yi et al. (2018) propose a model combining a spatial transformation component and a deep distributed fusion network to predict air quality in nine major cities in China. More recently, Shi et al. (2022) evaluate various deep learning models, including RNNs, LSTMs, GRUs, and Transformers, for air quality prediction in Beijing. Nationwide air quality forecasting in China has leveraged advanced architectures such as hierarchical group-aware graph neural networks (GAGNN) (Chen et al., 2023e), spatiotemporal graph neural networks (STGNNs) (Wang et al., 2020), and Transformer-based models (Liang et al., 2023; Yu et al., 2025). Additionally, RNNs have been utilized for air quality prediction in India (Arora et al., 2022) and Pakistan (Waseem et al., 2022), while hybrid CNN-LSTM architectures have been applied for predictions in Barcelona and Turkey (Gilik et al., 2022).

## 5.3 Sea Surface Temperatures

Variations in sea surface temperatures significantly influence El Niño–Southern Oscillation (ENSO) and La Niña events, which in turn have profound effects on global extreme climate conditions, such as increasing the likelihood of floods, droughts, heatwaves, and cold spells (Wang et al., 2023a). Niño 3.4 index, an important indicator for ENSO prediction, has been predicted using different deep learning (DL) models, such as RNN-based (Geng & Wang, 2021), CNN-based (Ham et al., 2019; Liu et al., 2021), residual CNNs (Hu et al., 2021b), ConvLSTM (He et al., 2019), GNN-based (Cachay et al., 2020), and Transformer-based models (Ye et al., 2021; Zhou & Zhang, 2023; Song et al., 2023). More recently, an adaptive graph spatial-temporal attention network (AGSTAN) has been proposed for longer lead (i.e., 23 months) ENSO prediction (Liang et al., 2024). Mu et al. (2021) evaluates multiple DL models for the Niño 3 index, Niño 3.4 index, and Niño 4 index with a multivariate air–sea coupler. Similar evaluation work involves comparing deep learning models for ENSO forecasting and presenting ENSO dataset (Mir et al., 2024). Moreover, some researchers directly predict the sea surface temperature using spatiotemporal graph attention networks (Gao et al., 2023c) and physical knowledge-enhanced generative adversarial networks (Meng et al., 2023). ENSO impacts have also been studied, including river flows (Liu et al., 2023b), rainfall (He et al., 2024b), and heatwaves (He et al., 2024a).

### 5.4 Flood

Accurate flood prediction is essential for mitigating the adverse impacts of flooding. Recent advances in deep learning (DL) have led to the development of various models tailored for flood forecasting and mapping, such as CNN-based (Adikari et al., 2021), RNN-based and LSTM (Nevo et al., 2022; Ruma et al., 2023), and CNN-RNN hybrid models such as ConvLSTM (Li et al., 2022), and LSTM-DeepLabv3+ (Situ et al., 2024a). Situ et al. (2024b) employs the *attention* mechanism for urban flood damage and risk assessment with improved flood prediction and land use segmentation. Furthermore, graph-based models have also gained attention for flood prediction (Kirschstein & Sun, 2024). FloodGNN-GRU combines GNNs and Gated Recurrent Units (GRUs) for spatiotemporal flood prediction by incorporating vector features like velocities (Kazadi et al., 2024) while Graph Transformer Network (FloodGTN) integrates GNNs and Transformers to learn spatiotemporal dependencies in water levels (Shi et al., 2023a) and the proposed `FIDLAr` (Shi et al., 2025) is used to mitigate floods. Additionally, physics-guided models further enhance flood prediction by embedding physical laws into model training. For instance, the DK-Diffusion model incorporates flood physics into its loss function to align predictions with hydrological principles (Shao et al., 2024). `DRUM` leverages diffusion model for operational flood forecasting and long-term risk assessment (Ou et al., 2024).

### 5.5 Drought

Drought, driven by a complex interplay of meteorological, agricultural, hydrological, and socio-economic factors, manifests across diverse spatial and temporal scales (Wilhite, 2016; Gyaneshwar et al., 2023). We focus on DL methods that consider meteorological drivers, such as precipitation deficits, wind patterns, and temperature anomalies, to predict various drought indices. LSTMs have been widely used to predict spatial precipitation patterns (dry-wet) (Gibson et al., 2021) and drought indices related to precipitation, such as the standardized precipitation index (SPI) (Poornima & Pushpalatha, 2019; Dikshit & Pradhan, 2021) and the standardized precipitation evapotranspiration index (SPEI) (Tian et al., 2021; Dikshit et al., 2021; Xu et al., 2022), excelling at capturing long-term dependencies. Beyond SPI and SPEI (Adikari et al., 2021; Dhyani & Pandya, 2021; Hao et al., 2023), CNNs have been applied for predicting other indices, such as the soil moisture index (SMI) (Dhyani & Pandya, 2021) and soil moisture condition index (SMCI) (Zhang et al., 2024b), aiding agricultural drought prediction. Hybrid models like ConvLSTM and CNN-LSTM have demonstrated significant improvements in multi-temporal predictions for SPEI (Danandeh Mehr et al., 2023; Nyamane et al., 2024) and SPI (Park et al., 2020), as well as indices like the scaled drought condition index (SDCI) (Park et al., 2020), composite drought index (CDI) (Zhang et al., 2023a), and Palmer drought severity index (PDSI) (Elbeltagi et al., 2024). Specifically, the CNN-GRU model has effectively forecasted daily reference evapotranspiration (ET) (Ahmed et al., 2022). Swin Transformer was used for drought prediction across multiple scales (Zhang et al., 2024a). Meanwhile, GANs have emerged as robust tools for drought prediction, with applications spanning vegetative drought prediction (Shukla & Pandya, 2023), and SMI (Ferchichi et al., 2024).

### 5.6 Tropical Storms/Cyclones and Hurricanes

Accurate forecasting of tropical storms, cyclones, and hurricanes is crucial for mitigating their devastating impacts. CNN-based models have been increasingly employed to predict various aspects of these phenomena, focusing on targets such as storm formation (Zhang et al., 2021; Nguyen & Kieu, 2024), intensity (Kim et al., 2024), track (Giffard-Roisin et al., 2020; Lian et al., 2020), and associated rainfall (Kim et al., 2022b). Hybrid models, such as CNN-LSTM, further improve the accuracy of intensity prediction (Alijoyo et al., 2024), extend lead times up to 60 hours (Kumar et al., 2022), and effectively capture landfall in terms of location and time (Kumar et al., 2021). GANs have also proven valuable in downscaling tropical cyclone rainfall to hazard-relevant spatial scales (Vosper et al., 2023) and in multitask frameworks for simultaneously forecasting cyclone paths and intensities (Wu et al., 2021). Recent approaches like diffusion models have been explored for forecasting cyclone trajectories and precipitation patterns (Nath et al., 2023). GNNs integrated with GRUs have been utilized to model storm surge dependencies across observation stations, offering improvements in spatial and temporal forecasting (Jiang et al., 2024).

## 5.7 Wildfire

Accurate wildfire prediction is critical for disaster management and mitigation. CNN-based models have demonstrated strong capabilities in wildfire spread prediction (Khennou et al., 2021; Shadrin et al., 2024), including forecasting fire weather with high spatial resolution (Son et al., 2022), generating spread maps (Huot et al., 2022), and modeling large-scale fire dynamics using multi-kernel architectures (Marjani & Mesgari, 2023). RNNs, including GRUs and LSTMs, excel in modeling wildfire risk and predicting spread, with GRU-LSTM showing superior performance in longer time series data (Perumal & Van Zyl, 2020; Dzulhijjah et al., 2023; Gopu et al., 2023). Hybrid CNN-LSTM models further enhance prediction accuracy, offering near-real-time daily wildfire spread forecasting (Marjani et al., 2024) and incorporating multi-temporal dynamics for prediction (Marjani et al., 2023). ConvLSTM models capture a wide range of temporal scales in wildfire prediction, from short-term intervals of 15 minutes (Burge et al., 2023) to longer-term forecasts extending up to 10 days (Masrur & Yu, 2023; Masrur et al., 2024). Other advancements include GANs, which have been utilized for wildfire risk prediction through conditional tabular data augmentation (Chowdhury et al., 2021), and GNNs, which simulate wildfire spread in variable-scale landscapes, effectively addressing landscape heterogeneity (Jiang et al., 2022). Additionally, researchers have also explored Transformer models for wildfire prediction (Miao et al., 2023; Cao et al., 2024).

# 6 Challenges and Future Directions

In this section, we introduce primary challenges and suggest promising future research opportunities from the perspectives of DL models (Subsections 6.1-6.3) and data (Subsections 6.3-6.4). While some of these challenges have been partially addressed for deterministic predictive models, it is essential to comprehensively characterize and revisit them in light of the recently emerging probabilistic generative and foundation models for weather forecasting.

## 6.1 Trustworthy AI

**Robustness:** Weather data is often subject to observational or collection biases, leading to significant performance degradation in data-driven models. These biases may stem from inconsistent data collection methods, non-uniformity or limited spatial or temporal coverage, and inaccuracies in sensor measurements. As a result, models trained on such biased data sets may struggle to generalize effectively. In addition to the bias issue, input instabilities (commonly referred to as adversarial vulnerabilities) where small, often imperceptible perturbations to the input can lead to disproportionately large changes in DL model predictions (Szegedy et al., 2013). `Opportunities:` (1) Bias correction with statistical adjustments (Durai & Bhradwaj, 2014) and data assimilation (Berry & Harlim, 2017; Qu et al., 2024b) can be applied to reduce biases in the data. (2) Adversarial training (Wang et al., 2024), a technique originally developed to defend against adversarial attacks in machine learning, can be adapted to weather prediction by exposing models to systematically perturbed data during training. This approach allows models to generalize better to biased or noisy inputs. Specifically, adversarial examples simulating systematic errors in weather observations can be incorporated alongside clean data to improve model robustness (Schmalfuss et al., 2023). Recent work on adversarial observations in weather forecasting (Imgrund et al., 2025) investigates how subtle perturbations in weather observations impact the outputs of DL models, offering valuable insights to in-domain applications. Shi et al. (2023b) added random noise to the observational data to mimic the input uncertainty and quantified the output uncertainty of DL models for other environmental applications. Despite such advances, research on effective remedies for adversarial vulnerabilities remains ongoing.

**Generalization:** AI models often fail to perform effectively on rare extreme weather or anomalous events that fall outside the distribution (OOD) of the training samples. `Opportunities:` (1) Physical laws represent precious wisdom from domain pioneers, but they are rarely explicitly incorporated into AI models (Feng et al., 2023). Leveraging physics-informed or physics-guided AI approaches can increase reliability and consistency with the physical world (Chen et al., 2021b; Meng et al., 2021; Yin et al., 2023), particularly while addressing extreme or unseen scenarios. Although significant progress has been made in the integration of physics and AI (see "Physics-AI" in Section 4), further exploration is needed to optimize and refine these

approaches. (2) DL models perform poorly in extreme weather events due to their rarity and limited representation in the training data. Effective data augmentation with generative diffusion models (Trabucco et al., 2023; Mardani et al., 2023) is a promising method to address or alleviate this challenge. By augmenting the training set with more extreme samples, DL models are better equipped to understand these rare events comprehensively, enhancing their generalizability. Therefore, it is worth exploring how to effectively augment data with extreme samples.

**Explainablity:** Neural networks are frequently referred to as "black boxes" due to the opacity of their internal processes, making it challenging to interpret how they produce outputs (Guidotti et al., 2018). In the weather and climate domains, understanding the underlying mechanisms of these models is of paramount importance and a necessity to ensure reliability and trustworthiness. `Opportunities:` Explainable AI tools, such as SHAP (Shapley Additive Explanations) (Lundberg, 2017), LIME (Local Interpretable Model-Agnostic Explanations) (Ribeiro et al., 2016), Grad-CAM (Selvaraju et al., 2017), and causal analysis (Zhang et al., 2011) have gained prominence in addressing this challenge. Furthermore, the principle of information bottleneck (IB) has been used for explainable learning in the time series domain (Feng et al., 2024a; Liu et al., 2024f). Given that weather data inherently constitute time series, we advocate exploring how the information bottleneck method can enhance the explainability of weather modeling. Leveraging these techniques can help determine whether DL models are producing meaningful results based on legitimate patterns or merely fabricating outputs, reinforcing trustworthiness and accountability in model predictions.

**Varying Resolution:** In weather and climate science, is it common to deal with varying data resolutions. For example, weather data have differing temporal and spatial resolutions across modalities. Meteorological observations might have an hourly temporal resolution from sparse sensors, radar echo data could feature six-minute temporal intervals and a spatial resolution of 1–4 km, and satellite imagery might exhibit a temporal resolution of 30 minutes with a spatial resolution of 5–12 km. These discrepancies complicate the task of harmonizing information across modalities for robust model development (Chen et al., 2023f). `Opportunities:` Therefore, an important challenge is to build models that can handle training data of varying resolutions and also reliably predict at a different resolution. Such models could revolutionize how we integrate data from various sources, including observations, satellite imagery, and numerical simulations, which often differ in granularity and format. `Aurora` processes input data with varying patch sizes (Bodnar et al., 2024), and `IPOT` (Inducing-point operator transformer) uses a smaller number of inducing points, flexibly handling any discretization formats of input (Lee & Oh, 2024).

**Uncertainty Quantification:** Given the chaotic nature of the atmosphere, quantifying uncertainty in weather predictions is essential to allow informed decision-making. Approaches such as initial conditions perturbation and Monte Carlo dropout have been studied (Bülte et al., 2024); however, they only simulate the aleatoric uncertainty, i.e., the inherent randomness in weather data or the epistemic uncertainty from the model itself due to limited knowledge. `Opportunities:` Generative diffusion models address both aleatoric and epistemic uncertainty simultaneously. Diffusion models learn the full probability distribution of the data, capturing aleatoric uncertainty through stochastic sampling, where the spread of outcomes reflects inherent data variability. When conditioned on the inputs, added stochastic noise incorporates input variability, further representing data-driven uncertainty. Furthermore, by initializing from different noise points, diffusion models capture epistemic uncertainty (Du & Li, 2023; Price et al., 2023), with greater variability in regions of sparse training data. This inherent stochasticity makes diffusion models a robust tool for quantifying both aleatoric and epistemic uncertainties.

### 6.2 Retrieval-augmented Foundation Models

Retrieval-augmented generation (RAG) (Gao et al., 2023a) has emerged as a promising approach to enhance foundation models by integrating external domain knowledge. `Opportunities:` While RAG has been extensively explored in domains such as medicine (Xiong et al., 2024), its application to weather and climate modeling remains underexplored. Depending on whether the foundation model uses diffusion models (Yang et al., 2023) or large language models (LLMs) (Zhao et al., 2023) as its underlying architecture, different opportunities arise for leveraging retrieval augmentation: (1) Diffusion Models for Weather Forecasting: In

the context of diffusion-based weather models (Shi et al., 2024), retrieval augmentation can be leveraged to fetch historical weather patterns similar to the current state, allowing it to recreate historical conditions that may have appeared in the past and that can serve as references to refine predictions, potentially improving accuracy and robustness (Liu et al., 2024a). RAG methods offer two key advantages for weather forecasting or more general time series prediction. First, it enables explicit access to relevant historical patterns during inference, allowing the model to leverage retrieved examples directly, rather than relying solely on information implicitly captured in model parameters. Second, RAG is particularly well-suited for improving performance on rare or extreme events, which are often underrepresented in training data and difficult for standard models to learn. By retrieving similar historical instances when such patterns reoccur, RAG enhances the model's ability to generalize and improves robustness in forecasting high-impact, low-frequency events. (2) LLMs for Weather Text Analysis: For tasks involving textual analysis of weather-related corpora, such as extreme weather reports or climatological summaries (Colverd et al., 2023), retrieval augmentation can provide valuable context by identifying and incorporating relevant documents. This approach can significantly enhance the model's ability to generate informed and contextually relevant outputs (Juhasz et al., 2024). By bridging retrieval-based methodologies with foundation models, RAG helps to maximize the power of foundation models, presenting an exciting avenue for advancing both accuracy and interpretability in weather and climate applications.

### 6.3 Multi-Modal Learning

Weather data comes from heterogeneous sources, encompassing observational sensors, radar, satellite imagery, and reanalysis data (Bai et al., 2022b; Lahat et al., 2015). For example, weather data has different temporal and spatial resolutions across modalities. Meteorological observations might have an hourly temporal resolution from sparse sensors, radar echo data could feature six-minute temporal intervals and a spatial resolution of 1–4 km, and satellite imagery might exhibit a half-hourly temporal resolution with a spatial resolution of 5–12 km. Furthermore, related weather data could be from unstructured textual information from expert reports and social media (Reichstein et al., 2019). These discrepancies complicate the task of harmonizing information across sources for robust model development (Chen et al., 2023f). `Opportunities:` A promising direction is to leverage multi-modal data to learn a joint representation of weather and climate events (Zhu et al., 2022). `CLLMate` (Li et al., 2024a) a recently emerged work, aligns numerical meteorological raster data with textual event data and leverages LLMs to predict weather and climate events, demonstrating how these two data modalities can effectively complement each other. Additionally, Qu et al. (2024a) develop a knowledge graph framework to automatically generate weather event overviews, supporting both prediction and reasoning tasks. Motivated by these advances, we believe that integrating knowledge graphs with numerical weather data presents a promising research direction for the weather forecasting domain.

### 6.4 Data Processing and Management

**Data Storage:** The volume of weather and climate data is increasing daily - European Centre for Medium-Range Weather Forecasts (ECMWF) archives contain about 450 PB of data to which 300 TB are added daily (Mukkavilli et al., 2023). `Opportunities:` Variational Autoencoder (VAE) approaches have emerged as powerful tools for data compression (Liu et al., 2024c; Han et al., 2024a), converting the high-dimensional data from the original space to a lower latent space. Liu et al. (2024c) reduce the data size from 8.61 TB to a compact 204 GB and Han et al. (2024a) compress the ERA5 dataset (226 TB) into a CRA5 dataset (0.7 TB). More importantly, they demonstrate that downstream experiments of global weather forecasting models trained on the compact CRA5 dataset achieve accuracy comparable to the models trained on the original dataset. This approach significantly reduces storage requirements for massive weather datasets.

**Data Quality:** The massive gridded reanalysis data are generated using mechanistic or statistical models that rely on empirical assumptions, raising concerns about the quality and reliability of the data. `Opportunities:` Data assimilation (Manshausen et al., 2024) is a promising method to increase data quality by calibrating model outputs with observational data, which could be remote sensing imagery and ground station measurements. For example, `SLAMS` proposes a conditional diffusion model to assimilate *in situ* weather station data and *ex situ* satellite imagery to effectively calibrate the vertical temperature profiles (Qu et al.,

2024b), and `ADAF` achieves effective data assimilation using real-world observations from different locations and multiple sources, including sparse surface weather observations and satellite imagery (Xiang et al., 2024). Furthermore, `EarthNet` is a multi-modal foundation model for global data assimilation of Earth observations utilizing masked autoencoders (Vandal et al., 2024). In summary, DL methods have become increasingly popular for integrating weather data from various sources to provide more precise representations.

### 6.5 Model Compression

While these large models (Bi et al., 2023; Lam et al., 2022; Price et al., 2023; Bodnar et al., 2024) yield impressive accuracies, they incur substantial training time and memory overhead, making them challenging to fine-tune or train from scratch (see Table 2). `Opportunities:` Model distillation is a well-established technique in which a smaller "student" model is trained to replicate the outputs or internal representations of a larger "teacher" model (Hinton et al., 2015; Xiang & Fujii, 2023). This approach significantly reduces computational resource needs in terms of processing and memory during inference while maintaining competitive predictive performance, as demonstrated by the recent success of DeepSeek (Guo et al., 2025). Given the substantial computational demands of large-scale weather forecasting models, applying distillation to this domain presents a promising avenue for exploration. Additionally, the architecture of selective state-space models (Mamba) (Gu & Dao, 2023) offers a more efficient alternative to transformers by capturing long-range dependencies with linear computational complexity. Although early efforts have begun to adapt Mamba for weather forecasting (Qin et al., 2024; Liu et al., 2024g), research in this area is still emerging.

### 6.6 Operational Deployment

**Interpretability and Accountability:** While these models offer enhanced predictive capabilities, their integration into decision-making pipelines raises important considerations around transparency and accountability, particularly for high-impact events such as hurricanes, floods, and heatwaves. The black-box nature of many ML models complicates their interpretability and challenges human forecasters' ability to verify or contest predictions, potentially undermining trust in automated weather forecasting systems. To mitigate risks, forecasting systems should incorporate robust uncertainty quantification, provide interpretable outputs, and be designed with human-in-the-loop oversight to ensure that domain experts can validate, override, or contextualize model outputs. Accountability mechanisms must also be established, including clear documentation of model updates, input data provenance, and version-controlled predictions to enable traceability and reproducibility. More importantly, hybrid models that integrate machine learning with physics-based approaches are essential for leveraging the complementary strengths of both paradigms.

**Ethical Consideration:** Equally important is the prevention of misuse. Advanced forecasts could be exploited for financial speculation, misinformation, or political manipulation, especially in sensitive contexts involving resource allocation or public warnings. As such, access to model outputs should be governed with careful consideration of use cases, and safeguards (e.g., delayed release of certain outputs, restricted access APIs) may be warranted in some operational settings. Adhering to established governance frameworks – such as the OECD AI Principles (OECD AI, 2019) and the UNESCO Recommendation on the Ethics of AI (UNESCO, 2021) – can help ensure that the deployment of AI-driven weather forecasting systems is ethically responsible and aligned with societal values.

## 7 Conclusions

In this work, we present a comprehensive and up-to-date survey of data-driven deep learning models and foundation models for weather prediction. We introduce a novel categorization of these models based on their training paradigms and provide an in-depth review, analysis, and comparison of key methodologies within each category. Additionally, we summarize available datasets, open-source codebases, and diverse real-world applications in a GitHub repository. More importantly, we present critical potential research directions for advancing AI-driven weather prediction, offering a roadmap for future research.

**Limitations.** This survey is particularly targeting the topic of weather prediction. The research topics in climate science are out of the scope, including climate downscaling (Ling et al., 2024a), climate emulation (Yu et al., 2024b), and climate trend prediction (Cael et al., 2023).

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

## Appendix

## A   Datasets

We summarize widely used benchmark datasets, where each data set is presented by domain, name, coverage, collection method, spatial and temporal resolution, time span, and the paper that introduces the dataset.

Table 3: Summary of Publicly Available Data Sets on Weather. CAM5: Community Atmospheric Model v5.

| Domain | Dataset | Coverage | Collect | Spatial | Temporal | Time Span | Paper |
|---|---|---|---|---|---|---|---|
| General Weather | WeatherBench | Global | Reanalysis | $1.40625°, 2.8125°, 5.625°$ | 6 hours | 1979-2018 | Rasp et al. (2020) |
| | WeatherBench 2 | Global | Reanalysis | $0.25°$ | 6 hours | 1979-2020 | Rasp et al. (2023) |
| | Weather2K | Region in China | Observation | - | 1 hour | 2017.01-2021.08 | Zhu et al. (2023) |
| | Weather5K | Global | Observation | - | 1 hour | 2014-2023 | Han et al. (2024b) |
| | HR-Extreme | Region in U.S. | Radar | 3 km×3 km | 1 hour | 2020-2020 | Ran et al. (2024) |
| Precipitation | SEVIR | Region in U.S. | Radar&Satellite | 1 km×1 km | 5 mins | 2017-2019 | Veillette et al. (2020) |
| | OPERA | Europe | Radar&Satellite | 2 km | 15 mins | 2019-2021 | Herruzo et al. (2021) |
| | Meteonet | France | Radar&Satellite | 1 km | 5-15 mins | 2016-2018 | Larvor et al. (2020) |
| | IMERG | Global | Radar&Satellite | 1 km | 30 mins | 2020-2023 | Huffman et al. (2020) |
| | HKO-7 | Region in Hong Kong | Radar | 1 km×1 km | 6 mins | 2009-2015 | Shi et al. (2017) |
| | Shanghai | Shanghai | Radar | 1 km | 6 mins | 2015-2018 | Chen et al. (2020) |
| | JMA | Japan | Radar | 1 km | 5 mins | 2015-2017 | Inoue & Misumi (2022) |
| | MRMS | CONUS and S. Canada | Radar | 1 km×1 km | 2 mins | 2017-2019 | Smith et al. (2016) |
| | RYDL | Germany | Radar | 1 km | 5 mins | 2014-2015 | Ayzel et al. (2020a) |
| | RainBench | - | | $5.625°$ | 1 hour | 2016-2019 | de Witt et al. (2021) |
| | IowaRain | Iowa, U.S. | Radar | 0.5 km×0.5 km | 5 mins | 2016-2019 | Sit et al. (2021) |
| | PostRainBench | Region in China | | 1 km×1 km | 3 hours | 2010-2021 | Tang et al. (2023) |
| Wind | GlobalWindTemp | Global | Observation | - | 1 hour | 2019-2010 | Wu et al. (2023) |
| | DigitalTyphoon | W.N. Pacific basin | Satellite | 5 km | 1 hour | 1978-2022 | Kitamoto et al. (2023) |
| | TropicalCyclone | Global | CAM5 simulation | 25 km | 3 hours | 1979-2005 | Racah et al. (2017) |
| | ClimateNet | Global | CAM5 simulation | 25 km | 3 hours | 1996-2010 | Kashinath et al. (2021) |
| Air Quality | UrbanAir | Regional, China | Observation | - | 1 hour | 2014-2015 | Zheng et al. (2013) |
| | KnowAir | Regional, China | Observation | - | 3 hours | 2015-2018 | Wang et al. (2020) |
| | ItalianAir | Italy | Observation | - | 1 hour | 2004-2005 | Vito (2016) |
| | BeijingAir1 | Regional, China | Observation | - | 1 hour | 2010-2014 | Chen (2017) |
| | BeijingAir2 | Regional, China | Observation | - | 1 hour | 2013-2017 | Chen (2019) |
| SST | OI SST v2 | Pacific Ocean | Observation&Satellite | $5°S-5°N, 170°W-120°W$ | Daily | 1982–2017 | Huang et al. (2019) |
| | ZonalWinds | Pacific Ocean | Reanalysis | $5°S-5°N, 120°E-160°E$ | Daily | 1982–2017 | Huang et al. (2019) |
| | TropicalOcean | Pacific Ocean | Observation | $5°S-5°N, 120°E-80°W$ | Monthly | 1982–2017 | Huang et al. (2019) |
| | SODA SST | Global | Reanalysis | $5° × 5°$ | Monthly | 1871–1973 | Geng & Wang (2021) |
| | GODAS | Global | Reanalysis | $5° × 5°$ | Monthly | 1994–2010 | Geng & Wang (2021) |
| | CMIP5 | Global | Simulation | $5° × 5°$ | Monthly | 1861–2004 | Geng & Wang (2021) |
| | ERA-Interim | Global | Reanalysis | - | Daily | 1984–2017 | Ham et al. (2019) |
| | CFSv2 | Global | Reanalysis | $5° × 5°$ | 6 hours | 1981–2017 | He et al. (2019) |
| | NOAA ERSSTv5 | Global | Observation | - | Monthly | 1854–2020 | Cachay et al. (2020) |
| | CMIP6 | Tropical Pacific | Simulation | $2° × 0.5°$ | Monthly | 1850–2014 | Zhou & Zhang (2023) |
| | ORAS5 | Tropical Pacific | Reanalysis | - | Monthly | 1958–1979 | Zhou & Zhang (2023) |
| | NOAA/CIRE | Global | Reanalysis | $2° × 2°$ | 6 hours | 1850–2015 | Mu et al. (2021) |
| | REMSS | Global | Satellite | $0.25° × 0.25°$ | Daily | 1997–2020 | Mu et al. (2021) |
| | ENSO | Tropical Pacific | NOAA, NCEI, NCAR | - | Monthly | 1950–2023 | Mir et al. (2024) |
| | GHRSST | South China Sea | Observation | $1.20° × 1.20°$ | Daily | 2007–2014 | Meng et al. (2023) |
| | HYCOM | South China Sea | Simulation | $1.12° × 1.12°$ | Daily | 2007–2014 | Meng et al. (2023) |
| | Hadley-OI SST | Global | Observation&Satellite | $1° × 1°$ | Monthly | 1870–2020 | Liu et al. (2023b) |
| | COBE SST | Global | Observation | $1° × 1°$ | Monthly | 1891–2020 | Liu et al. (2023b) |
| | SILO SST | Australia | Observation | - | Monthly | 1921–2020 | He et al. (2024b) |
| | OISST | Global | Observation&Reanalysis | $0.25° × 0.25°$ | Daily | 1982–2020 | He et al. (2024a) |
| | ERA5 | Global | Observation&Reanalysis | $0.25° × 0.25°$ | 1 hour | 1982–2020 | He et al. (2024a) |
| Flood | DEM | Carlisle, UK | Observation | 5 m | 1 hour | 2005-2015 | Kabir et al. (2020) |
| | AustraliaFlood | Australia | Observation | - | Daily | 1900-2018 | Adikari et al. (2021) |
| | SekongFlood | Vietnam, Laos, Cambodia | Observation | - | Daily | 1981-2013 | Adikari et al. (2021) |
| | BangladeshFlood | Bangladesh (GBM river network) | Observation | - | Daily | 1979-2014 | Ruma et al. (2023) |
| | GermanyFlood | Germany, Sachsen | Radar | 1 km | 1 hour | Different periods | Li et al. (2022) |
| | ElbeRiverFlow | Germany, Elbe River in Sachsen | Observation | - | 1 hour | Different periods | Li et al. (2022) |
| | FijiFlood | Fiji Islands | Observation | - | Daily | 1990-2019 | Moishin et al. (2021) |
| | FloridaFlood | USA, Coastal South Florida | Observation | - | 1 hour | 2010-2020 | Shi et al. (2025) |
| | QijiangRiverBasin | China, Chongqing, Qijiang River | Observation | - | 1 hour | 1979-2020 | Shao et al. (2024) |
| | TunxiRiverBasin | China, Anhui, Tunxi River | Observation | - | 1 hour | 1981-2007 | Shao et al. (2024) |
| Drought | MODIS | Regional, China | Satellite | 500 m | Monthly | 2000-2020 | Zhang et al. (2023a) |
| | CHIRPS | Regional, China | Satellite | $0.05°$ | Monthly | 2000-2020 | Zhang et al. (2023a) |
| | ChinaDrought | China | - | - | Monthly | 1980-2019 | Xu et al. (2022) |
| | IndianDrought | Peninsular, India | Satellite | $0.25° × 0.25°$ | Daily | 1981-2021 | Shukla & Pandya (2023) |
| | AVHRR | Peninsular, India | Radiometer | 1 km | Daily | 1981-2022 | Shukla & Pandya (2023) |
| | ERA5 | East Asia | Reanalysis | $0.25° × 0.25°$ | 1 hour | 1970-2020 | Zhang et al. (2024a) |
| | EastAsiaDrought1 | East Asia | Satellite | $0.25°$ | Daily | 2003-2018 | Park et al. (2020) |
| | EastAsiaDrought2 | East Asia | Satellite | $0.05°$ | 16 days | 2003-2018 | Park et al. (2020) |
| | EastAsiaDrought3 | East Asia | Satellite | $0.05°$ | 8 days | 2003-2018 | Park et al. (2020) |
| | EastAsiaDrought4 | East Asia | Simulation | $0.5°$ | 3 hours | 2015-2018 | Park et al. (2020) |
| | EastAsiaDrought5 | East Asia | Satellite | 90 m | - | - | Park et al. (2020) |
| | EastAsiaDrought6 | East Asia | Satellite | $0.5°$ | Yearly | - | Park et al. (2020) |
| Wildfire | LANDFIRE PROGRAM | California | Satellite | $128 × 128$ | 15 mins | - | Burge et al. (2023) |
| | FARSITE | Regional | Synthetic | 30 m | 15 mins | - | Burge et al. (2023) |
| | NASA-MODIS Terra | California | Satellite | 1 km | 5 mins | 2017-2018 | Chowdhury et al. (2021) |
| | MERRA-2 | California | Reanalysis | $0.5° × 0.625°$ | 1 hour | 2017-2018 | Chowdhury et al. (2021) |
| | USGS | Regional | Satellite | 30 m | - | 2017-2018 | Chowdhury et al. (2021) |
| | AICC | Regional, Alaska | Satellite | $400 × 350$ | Daily | 2002-2018 | Marjani et al. (2023) |
| | NRC | Regional, Canada | Satellite | 30 m | Daily | 2002-2018 | Marjani et al. (2023) |
| | VIIRS | South Africa | Satellite | 375 m | 1 hour | 2012-2014 | Perumal & Van Zyl (2020) |
| | VIIRS | California | Satellite | 375 m | Daily | 2012-2021 | Masrur et al. (2024) |
| | Percolation model | Regional | Synthetic | $110 × 110$ | 5 mins | - | Masrur et al. (2024) |

# B  Model Architectures

## B.1  Convolutional Neural Networks

Convolutional Neural Networks (CNNs) LeCun et al. (1995) are a specialized type of neural network designed for processing structured grid data, such as images. The convolutional layer usually utilizes convolutional kernels to process the input data, performing convolution operations to extract features like edges, textures, and patterns Li et al. (2021). This is often followed by a pooling layer to reduce the spatial dimensions of the feature maps, making the network computationally more efficient and focusing on the most important information.

They are widely used in tasks related to computer vision, such as image classification He et al. (2016), object detection Ren et al. (2016), and segmentation He et al. (2017). Moreover, CNNs could be categorized into Conv1D, Conv2D, and Conv3D according to the sliding dimension of convolutional kernels Kiranyaz et al. (2021).

## B.2  Recurrent Neural Networks

Recurrent Neural Networks (RNNs) Medsker & Jain (2001) is a type of neural network particularly suited for tasks involving time-dependent or sequential data, such as time series forecasting Sbrana et al. (2020), natural language processing Mikolov et al. (2011); Zhang et al. (2017), and speech recognition Yadav et al. (2022). The key idea behind this is to recurrently learn from a sequence of data with an internal (hidden) state, which includes as inputs the previous hidden states and current input. The learning or update rule is:

$$
\begin{aligned}
h_t &= \sigma(\mathbf{W}_x x_t + \mathbf{W}_h h_{t-1} + b_h), \\
y_t &= \sigma(\mathbf{W}_y h_t + b_y),
\end{aligned}
\tag{1}
$$

where $h_t$ is the hidden state at $t$-th time step, $x_t$ is the input at $t$-th time step, $y_t$ is the output at the same time step, $\mathbf{W}_x$, $\mathbf{W}_h$, and $\mathbf{W}_y$ are the weight matrices, $b_h$ and $b_y$ are the biases, and $\sigma$ is the activation function (e.g., tanh or ReLU).

However, RNNs often suffer from gradient vanishing and gradient explosion while modeling long sequences. Long Short-Term Memory Hochreiter & Schmidhuber (1997) (LSTM) and Gated Recurrent Unit Chung et al. (2014) (GRU) have been proposed to alleviate such a problem by well-designed gates to forget and filter information.

## B.3  Graph Neural Networks

Graph Neural Networks (GNNs) Scarselli et al. (2008) is designed to work on graph-structured data, $\mathcal{G} = (\mathcal{V}, \mathcal{E})$, consisting of a set of nodes $\mathcal{V}$ and a set of edges $\mathcal{E}$. These nodes and edges represent the entities and the dependent relationships among these entities, respectively. Spatio-temporal Graph Neural Networks (ST-GNNs) Yu et al. (2017) is an extension of GNNs designed to model both spatial and temporal dependencies in dynamic graph-structured data changing over time, $\mathcal{G}_t = (\mathcal{V}, \mathcal{E}, t)$. Here, nodes $\mathcal{V}$ refer to spatial locations, and edges $\mathcal{E}$ refer to spatial relationships. Each node $v_t^i$ represents the feature vector at the corresponding location $i$ and time $t$. For each node, the message-passing technique Gilmer et al. (2017) is often employed to capture the spatial dependencies on its neighbors. The temporal dependencies between graph snapshots can be modeled with the sequential models aforementioned. For the message passing, hidden states $h_t^i$ at each node are updated based on messages (feature vectors) $v_{t+1}^i$ according to:

$$
\begin{aligned}
v_{t+1}^i &= \sum_{j \in N(i)} M_t(h_t^i, h_t^j, e_{ij}), \\
h_{t+1}^i &= \sigma(h_t^i, v_{t+1}^i),
\end{aligned}
\tag{2}
$$

where in the sum, $N(i)$ denotes the neighbors of $i^{th}$ node in graph $\mathcal{G}$. After iterative updates $k$ time steps, the final output of the whole graph at time $t + k$ can be computed with a readout function $\mathcal{O}$:

$$
y_{t+k} = \mathcal{O}(\{h_{t+k}^i \mid i \in \mathcal{G}\}).
\tag{3}
$$

## B.4 Transformer and Vision Transformer

To overcome the limitations of RNNs, which stem from their inherent sequential processing, the Transformer model Vaswani (2017) has emerged as a powerful alternative. Its core innovation lies in the use of parallel processing through the *attention* mechanism, enabling it to capture dependencies between any parts of a sequence without the need for sequential steps Wen et al. (2022). The *attention* mechanism is described as follows:

$$\text{Attention}(\mathbf{Q}, \mathbf{K}, \mathbf{V}) = \text{softmax}\left(\frac{\mathbf{Q}\mathbf{K}^T}{\sqrt{d_k}}\right)\mathbf{V}, \tag{4}$$

where the $d_k$ denotes the dimension of the key, $\mathbf{Q} \in \mathbb{R}^{n \times d_k}$, $\mathbf{K} \in \mathbb{R}^{m \times d_k}$, and $\mathbf{V} \in \mathbb{R}^{m \times d_v}$ are the query matrix, key matrix, and value matrix, respectively. These three matrices are computed by linear transformations from the original input sequence $\mathbf{X} \in \mathbb{R}^{n \times d}$ with learnable weight matrices $\mathbf{W}_q \in \mathbb{R}^{d \times d_k}$, $\mathbf{W}_k \in \mathbb{R}^{d \times d_k}$, $\mathbf{W}_v \in \mathbb{R}^{d \times d_v}$, as

$$\mathbf{Q} = \mathbf{X}\mathbf{W}_q, \mathbf{K} = \mathbf{X}\mathbf{W}_k, \mathbf{V} = \mathbf{X}\mathbf{W}_v. \tag{5}$$

**Vision Transformer.** The Vanilla Transformer was originally proposed for dealing with sequences. Vision Transformer (ViT) Dosovitskiy et al. (2020) is a variant tailed to process images and has shown powerful performance compared to convolutional neural networks (CNNs). ViT models divide the input image into a grid of smaller, non-overlapping patches. Each patch is treated similarly to a "word" in natural language processing, and the patches are then flattened into vectors. Positional embeddings are added to these patch embeddings to mark the relative positions of patches in the image, helping models understand the image's spatial layout. Subsequently, the additive embeddings are fed into the Vanilla Transformer layer to leverage the *attention* mechanism. We refer readers to look into Figure 1 in Dosovitskiy et al. (2020).

## B.5 Mamba and Vision Mamba

We start by introducing the State Space Models (SSMs). SSMs represent the evolution of the system's internal states and make predictions of what their next state could be. For sequence modeling, SSMs map a sequence $x(t) \in \mathbb{R}^L \mapsto y(t) \in \mathbb{R}^L$ through an implicit latent state $h(t) \in \mathbb{R}^{L \times N}$:

$$\begin{aligned} h'(t) &= \mathbf{A}h(t) + \mathbf{B}x(t), \\ y(t) &= \mathbf{C}h(t), \end{aligned} \tag{6}$$

where $\mathbf{A} \in \mathbb{R}^{N \times N}$ and $\mathbf{B}, \mathbf{C} \in \mathbb{R}^{N \times 1}$ are learnable matrices. The continuous sequence is discretized by a step size $\Delta$, and the discretized SSM model is represented as:

$$\begin{aligned} h_t &= \bar{\mathbf{A}}h_{t-1} + \bar{\mathbf{B}}x_t, \\ y_t &= \mathbf{C}h_t, \end{aligned} \tag{7}$$

where discretization rule can be achieved by zero-order hold Zhang & Chong (2007) $\bar{\mathbf{A}} = \exp(\Delta\mathbf{A})$ and $\bar{\mathbf{B}} = (\Delta\mathbf{A})^{-1}(\exp(\Delta\mathbf{A}) - \mathbf{I}) \cdot \Delta\mathbf{B}$. The structured state-space model (S4), a variant of the vanilla SSM, improves long-range dependency modeling by utilizing the High-order Polynomial Projection Operators (HiPPO) Gu et al. (2020).

**Mamba.** S4 applies the same parameters $\mathbf{A}$ and $\mathbf{B}$ to each "token" of input, which is challenging to identify the importance of each input. Selective State Space Model (Mamba) Gu & Dao (2023) incorporates a selection mechanism such that parameters that affect interactions along the sequence are input-dependent (parameters $\Delta$, $\mathbf{A}$, $\mathbf{B}$ are functions of the input), enabling capturing contextual information in long sequences. Besides, Mamba possesses efficient hardware-aware designs. It utilizes three computing acceleration techniques (kernel fusion, parallel scan, and recomputation) to materialize the hidden state $h$ only in more efficient levels of the GPU memory hierarchy.

**Vision Mamba.** Vision Mamba Zhu et al. (2024a) is a variant of Mamba used for image modeling. Similar to Vision Transformer, Vision Mamba first splits the input image into patches and then projects them into

patch tokens, but leverages bidirectional SSMs (Mamba blocks) to replace attention mechanisms as the image encoder to model the sequence of tokens. Therefore, Vision Mamba can be well-tailed for 2-D grid weather data, e.g., MetMamba Qin et al. (2024).

## B.6  Generative Adversarial Networks

Generative Adversarial Networks (GANs) Goodfellow et al. (2014); Mirza (2014) were originally proposed to learn a generative model to generate realistic images via adversarial training. Specifically, GANs simultaneously train two neural networks adversarially: a `Generator G` and a `Discriminator D`. The Generator learns the underlying data distribution and generates produce samples that can effectively fool the discriminator, while the discriminator differentiates between the samples generated by the generator and the real samples by outputting the corresponding probabilities. This training process can be regarded as a two-player zero-sum game Washburn & Wood (1995), ultimately ending when the discriminator is unable to distinguish between the generator-generated samples and the real samples, i.e., $D(x) = \frac{1}{2}$.

GANs have widely used for image generation Xu et al. (2018), super-resolution Harder et al. (2022), style transferring Zheng et al. (2022), and image-based weather forecasting Chen et al. (2022); Choi et al. (2023); Cheng et al. (2023).

## B.7  Diffusion Models

Diffusion Models (DMs) Ho et al. (2020); Song et al. (2020) are the other type of generative models that have gained significant popularity in computer vision Saharia et al. (2022); Croitoru et al. (2023), natural language processing Hertz et al. (2022); Li et al. (2023b), due to their ability to produce high-quality, realistic samples. Diffusion models work in two processes: *forward diffusion process* and *reverse denoising process*. In the forward process, data (e.g., an image) is gradually "noised" by adding small amounts of Gaussian noise over multiple steps until it becomes nearly pure noise. This process is usually fixed and non-learnable, where each step incrementally increases the noise. The reverse process is learnable, where the model learns how to gradually remove noise, step-by-step, to recover a realistic sample from a noisy starting point. This iterative denoising process helps to learn the intricate, high-dimensional data distribution.

Mathematically, the *forward process* transforms an input $\mathbf{x}_0$ with a data distribution of $q(\mathbf{x}_0)$ to a white Gaussian noise vector $\mathbf{x}_N$ in $N$ diffusion steps. It can be described as a Markov chain that gradually adds Gaussian noise to the input according to a variance schedule $\{\beta_1, \ldots, \beta_N\} \in (0, 1)$:

$$q(\mathbf{x}_{1:N} \mid \mathbf{x}_0) = \prod_{n=1}^{N} q(\mathbf{x}_n \mid \mathbf{x}_{n-1}), \tag{8}$$

where at each step $n \in [1, N]$, the diffused sample $\mathbf{x}_n$ is obtained with $q(\mathbf{x}_n \mid \mathbf{x}_{n-1}) = \mathcal{N}\left(\mathbf{x}_n; \sqrt{1 - \beta_n}\mathbf{x}_{n-1}, \beta_n\mathbf{I}\right)$.

In the *reverse process*, the *denoiser network*, $p_\theta(\cdot)$, is used to recover $\mathbf{x}_0$ by gradually denoising $\mathbf{x}_n$ starting from a Gaussian noise $\mathbf{x}_N$ sampled from $\mathcal{N}(0, \mathbf{I})$. This process is presented as:

$$p_\theta(\mathbf{x}_{0:N}) = p(\mathbf{x}_N) \prod_{n=1}^{N} p_\theta(\mathbf{x}_{n-1} \mid \mathbf{x}_n). \tag{9}$$

In weather and climate domains, diffusion models have been applied to precipitation nowcasting Asperti et al. (2023a); Gao et al. (2024), atmospheric downscaling Ling et al. (2024a); Mardani et al. (2023), weather forecasting Shi et al. (2024); Andrae et al. (2024).

