# OpenReview forum: "Deep Learning and Foundation Models for Weather Prediction: A Survey"
_TMLR — Rejected by TMLR_

### Review · Reviewer_p1vr · 2025-02-11

**Summary Of Contributions:**

This paper presents a comprehensive survey of deep learning (DL) and foundation models for weather prediction, offering a novel taxonomy based on training paradigms: deterministic predictive learning, probabilistic generative learning, and pre-training and fine-tuning. The authors provide a detailed overview of state-of-the-art models, analyze their strengths and limitations, and compile an extensive repository of resources, including benchmark datasets, open-source codes, and real-world applications. Additionally, the paper outlines ten critical research directions across five key avenues to advance DL methods for weather prediction. The work is well-structured, thorough, and timely, given the increasing importance of accurate weather forecasting in the context of climate change.

**Audience:**

Yes

**Broader Impact Concerns:**

The paper highlights the potential benefits of improved weather prediction, such as better disaster preparedness and mitigation of climate change impacts. However, it should also address potential ethical concerns, such as the misuse of advanced forecasting technologies for malicious purposes or the unequal distribution of benefits across different regions and communities. Additionally, the paper could discuss the environmental impact of training large DL models, particularly in terms of energy consumption and carbon footprint.

Overall, this is a well-written and comprehensive survey that makes a significant contribution to the field of weather prediction. With the suggested minor changes, it will be an even stronger and more impactful work.

**Claims And Evidence:**

Yes

**Requested Changes:**

1. **Expand on Real-World Deployment**: Include a section or subsection discussing the practical challenges of deploying DL models in operational weather forecasting systems, including computational constraints, data availability, and integration with existing infrastructure.
2. **Comparative Analysis**: Add a comparative analysis of the performance of different models across various tasks and datasets, possibly in the form of a table or summary figure, to provide a clearer picture of the strengths and weaknesses of each approach.
3. **Ethical Considerations**: Expand the discussion on broader impact concerns, particularly focusing on ethical considerations such as the potential misuse of advanced forecasting technologies and the societal implications of improved weather prediction.

**Strengths And Weaknesses:**

### Strengths:
1. **Novel Taxonomy**: The introduction of a systematic categorization based on training paradigms is a significant contribution, providing a clear framework for understanding the diverse landscape of DL models in weather prediction.
2. **Comprehensive Overview**: The paper offers an extensive survey of state-of-the-art models, covering a wide range of architectures and applications, which will be highly valuable for researchers and practitioners in the field.
3. **Resource Compilation**: The curated summary of open-source code repositories and widely used datasets is a practical resource that bridges research advancements with real-world implementations.
4. **Future Directions**: The paper not only reviews existing work but also provides a forward-looking roadmap, highlighting critical research directions that will guide future work in the field.

### Weaknesses:
1. **Limited Discussion on Real-World Deployment**: While the paper discusses theoretical and methodological advancements, it could benefit from a deeper discussion on the challenges and opportunities of deploying these models in real-world operational settings.
2. **Lack of Comparative Analysis**: Although the paper provides a taxonomy and overview of models, a more detailed comparative analysis of the performance of different models across various tasks and datasets would strengthen the paper.
3. **Broader Impact Concerns**: The paper briefly mentions the societal implications of improved weather prediction but does not delve deeply into ethical considerations, such as the potential misuse of advanced forecasting technologies.

---

> ### Author Response · Authors · 2025-04-27
> **Author Response**
>
> We greatly appreciate your acknowledgments and suggestions. We provide our responses as follows:
>
> >**Q1:** Deeper discussion on the challenges and opportunities of deploying these models in real-world operational settings.
>
> We have expanded the discussion on possible challenges and opportunities in real-world operational deployments. Specifically, we highlighted them from the following perspectives:  **Interpretability, Accountability, and Integration into Existing Systems**. Please see Section 6.6.
>
>
> >**Q2:** More detailed comparative analysis of the performance of different models across various tasks and datasets would strengthen the paper.
>
> In Table 2, we added more comparative analysis of the performance of different models across various tasks and datasets.
>
> >**Q3:** Delve deeply into ethical considerations, such as the potential misuse of advanced forecasting technologies.
>
> Included the discussion on broader ethical concerns. Please check out Section 6.6.

---

### Review · Reviewer_1bNc · 2025-03-12

**Summary Of Contributions:**

This manuscript is a survey paper of deep learning weather prediction models that provides a comprehensive list of available models, and proposes a taxonomy (a categorization) of the models into three broad classes: deterministic predictive learning, probabilistic generative learning, and pre-training and fine-tuning learning (also known as foundation models). These approaches are each summarized and compared.

**Audience:**

Yes

**Claims And Evidence:**

Yes

**Requested Changes:**

I have a few suggestions for clarification and improvement:

- In abstract: the authors say these prediction models "surpass" traditional physics-based methods, but I believe this statement can be qualified a bit, since the situation is more subtle. I believe the authors know this too, I just am of the opinion a balanced statement saying they surpass it for some given benchmark problems would be more informative. (e.g. for extrapolation numerical simulation is more reliable).

- page 2, paragraph 2: "their predictions are often blurry" : How blurry? There is some detail in Table 1, however this table is not explained.

- page 3: station-based observation data vs. gridded reanalysis data : station-based observation data seems to refer to measurement data from locations where real measurement stations are available whereas gridded reanalysis data comes from synthetic simulation data. This distinction  might clarify this categorization.

- page 4: Definition 2.1-3. These definitions are not mathematical definitions, and I would suggest avoiding using the definition environment.

- page 5: Figure 3 is very informative. I believe this figure can appear earlier to give the readers the taxomony at first glance.

- page 5: Table 1: As mentioned above, this table can be explained a little more detail.

- page 6: paragraph 3: it is not clear to me what "the greedy algorithm is used to guarantee the minimal number of iterations" means, a clarification would be nice.

**Strengths And Weaknesses:**

Strengths

The manuscript provides a substantial list of models that are proposed, and this reviewer is not aware of well-known methods outside of the ones mentioned in the manuscript.  The comparison of models are concise and insightful.  It can be a very useful resource for new researchers entering the field of deep learning for weather prediction, or practitioners seeking to make use of the state-of-the-art prediction models.

Weaknesses

The survey article is certainly useful to the community, but by nature does not contain and new results. It is not clear to me whether this venue (TMLR) is appropriate for such a manuscript, since it does not have correct new results that are not necessarily of high impact. This is not a criticism of the quality of the manuscript, however, and I leave to the editor whether this is relevant for TMLR.

---

> ### Author Response · Authors · 2025-04-27
> **Author Response**
>
> > **Q1:** The survey article is certainly useful to the community, but by nature does not contain and new results. It is not clear to me whether this venue (TMLR) is appropriate for such a manuscript, since it does not have correct new results that are not necessarily of high impact. This is not a criticism of the quality of the manuscript, however, and I leave to the editor whether this is relevant for TMLR.
>
> We thank the reviewer for recognizing the quality of this work. TMLR indeed solicits and accepts “survey papers”, as mentioned in the website: [https://jmlr.org/tmlr/papers/](https://jmlr.org/tmlr/papers/).
>
> > **Q2:** In abstract: the authors say these prediction models "surpass" traditional physics-based methods, but I believe this statement can be qualified a bit, since the situation is more subtle. I believe the authors know this too, I just am of the opinion a balanced statement saying they surpass it for some given benchmark problems would be more informative. (e.g. for extrapolation numerical simulation is more reliable).
>
> Rewrote the abstract and the related sentences in the introduction (see changes just below Figure 1).
>
> > **Q3:** page 2, paragraph 2: "their predictions are often blurry" : How blurry? There is some detail in Table 1, however this table is not explained.
>
> At the beginning of Section 4, we provided some comparisons and added more explanations for Table 1.
>
> >  **Q4:** page 3: station-based observation data vs. gridded reanalysis data : station-based observation data seems to refer to measurement data from locations where real measurement stations are available whereas gridded reanalysis data comes from synthetic simulation data. This distinction might clarify this categorization.
>
> Rewrote it. Please see changes (marked in blue) in Section 3.1.
>
> > **Q5:** page 4: Definition 2.1-3. These definitions are not mathematical definitions, and I would suggest avoiding using the definition environment.
>
> Reformatted them.
>
> >  **Q6:** page 5: Figure 3 is very informative. I believe this figure can appear earlier to give the readers the taxonomy at first glance.
>
> We moved it to the beginning of the second page.
>
> > **Q7:** page 5: Table 1: As mentioned above, this table can be explained a little more detail.
>
> Added more detailed explanations for Table 1.
>
> > **Q8:** page 6: paragraph 3: it is not clear to me what "the greedy algorithm is used to guarantee the minimal number of iterations" means, a clarification would be nice.
>
> It appears in the Pangu-Weather work. They train four individual models for lead times of 1, 3, 6, and 24 hours. In the testing stage, given a forecast goal with a certain lead time, the greedy algorithm is used to guarantee the minimal number of iterations of the trained models for that forecast window. **For example**, for a 7-day forecast, Pangu executes a 24-hour forecast 7 times; while for a 23-hour forecast, Pangu executes a 6-hour forecast 3 times, followed by a 3-hour forecast 1 time and a 1-hour forecast 2 times. We also added this example in the manuscript for clarification (see page 6, paragraph 2).

---

> ### Comment · Reviewer_1bNc · 2025-05-09
>
> I thank the authors for addressing the issues I have raised, the changes seem reasonable.
>
> Since this is a survey article, I would like to ask the authors if they considered the implication of the known input instabilities in deep learning models, in the name of "adversarial examples" (Szegedy et al, "Intriguing Properties of Neural Networks"). An important downside of deep learning prediction models (vs. numerical simulations) is there is no known remedy for these input instabilities and the prediction results can vary wildly if the input is perturbed slightly. It is this reviewer's opinion that this is a critical issue for high-consequence applications like weather prediction. If the authors agree, I would recommend at least mentioning this persistent problem in the survey.

---

> > ### Author Response · Authors · 2025-05-12
> >
> > Thanks for raising the great suggestion. We have now extended Section 6.1 in the revised manuscript to discuss it further.

---

### Review · Reviewer_YLtG · 2025-04-14

**Summary Of Contributions:**

The present paper is a survey, TMLR declares *surveys that draw new connections, highlight trends, and suggest new problems in an area.* are within scope. Hence, I will center my review around these three criteria.

The work presents a vast list of literature broadly related to numerical weather prediction (NWP) with deep neural networks. This vast body of literature is structure by the methodology used and by the application considered. This includes applications beyond typical weather forecasting tasks, such as air quality, drought, flood and wild fire prediction. The work also introduces a Markdown file in a Github repo, which lists all of the cited works. Finally, ten broad research topics are identified which the authors propose as interesting for future work.

**Audience:**

No

**Claims And Evidence:**

No

**Requested Changes:**

- Introduce a proper systematic methodology for the literature review, e.g. by querying existing literature databases with keyword search or document embeddings.
- Add missing papers (e.g. AtmoRep)
- Restructure the categorization and rewrite the article, such that new connections between the surveyed articles become evident
- Properly explain terms related to NWP
- Properly explain how the mentioned applications relate to NWP
- Remove generic points from the presented future directions
- Introduce new future directions, which are not yet well-studied in current literature, but rather arise from problems of current approaches - from conventional NWP and also from AI-based NWP
- Remove the points related to GenAI with weather constraints
- Develop a key message for this survey
- Rewrite the abstract and introduction to be less verbose, but rather clearly display the key message of this survey
- Remove the section in the introduction that focusses on other surveys
- Remove the contributions in the introduction
- Add proper explanation of the different approaches of conventional NWP. Especially make clear the differences and commonalities between (radar) nowcasting, medium-range weather forecasting and seasonal prediction.
- Please note: for most weather prediction tasks, it is enough to look at benchmarks to understand which methods are SOTA. Hence, this alone is not a meaningful contribution of a survey.

**Strengths And Weaknesses:**

### Strengths:
1. The authors have collected a large pile of literature, and identified some commonalities between papers, which may enable readers to identify similar works
2. The fact that most cited papers are fairly recent shows that this survey highlights a trend

### Weaknesses:
1. The literature review is not systematic and misses out on relevant publications
2. The categorization of papers into bins of methodologies is at best loosely applicable, and more importantly, it is not clear how this categorization draws new connections or is otherwise particularly useful
3. Large parts of the paper are short 1-sentence summaries of individual papers with little to no narrative behind. This hampers the readability of the paper and makes it hard to identify the core message
4. The presentation and use of terms and ideas from meteorology and Earth system science ranges from poor to outright wrong.
5. The presented future research directions are either extremely general and apply to almost all applications of machine learning (robustness, generalization, explainability, uncertainty quantification and data quality), suggest using a new development from mainstream ML also for weather forecasting (RAG, Multi-modal learning), have already been published quite a lot on weather forecasting (data storage/compresison, varying resolution) or are not properly explained and seem to misinterpret findings from meteorology and geography  (GenAI with weather constraints).


I will provide a non-exhaustive list of further comments and examples below.
- An example on the overly excessive use of verbosity in this text. The content presented in the first 72 words of the abstracts can be easily said with less than a third of the words....: "Conventional numerical weather prediction models are computationally expensive. Deep learning often offers faster, and sometimes better predictions but still faces key challenges."
- the use of `real-world applications` in the abstract makes no sense, NWP itself is a real-world application.
- Motivating this survey with the issue of climate change seems not straight-forward, NWP is important regardless of climate change, I'd rather suggest to motivate why AI seems a good idea for NWP.
- The first two points on why conventional numerical solvers are hampered for NWP (compute cost scales with resolution & parametrizations of subgridscale processes are uncertain) need more elaborate explanation. Currently the points are only understandable if readers are already aware of them. In addition, both points could use a few more references.
- The third point is wrong: Current operational NWP systems are run in large ensembles to produce probabilistic weather forecasts. In fact, it is rather that most ML-based approaches (at least until ~2 years ago) have fallen short here, by just focussing on deterministic forecasting.
- Introducing ARIMA immediately after conventional NWP systems seems unlogical. ARIMA and other time series forecasting methods are only seldom used (most weather data is spatio-temporal) and only for tasks like ENSO prediction (which is correctly identified).
- The Introduction discusses the advancements in AI-based NWP a bit to enthusiastically. For example, while PanguWeather and GraphCast are surely important contributions, it is less easy to say the outperform e.g. IFS already, as they did not produce probabilistic forecasts. Also the comparison in compute costs needs to be more nuanced, most AI-based NWP runs on GPUs while most conventional solvers run on CPUs.
- Section 2.1 needs to explain clearly what reanalysis data is, and what original data sources are used to produce a reanalysis. Also the section totally misses out on radar, satellite, balloon and aircraft data, which are arguably at least as important as station data.
- Section 2.2 leaves the reader with a false impression of NWP. Only very rarely are forecasts made specifically for particular extreme events. Rather NWP forecasts are produced, and afterwards analyzed for the prevalence of extremes.
- Equation 1 brings no benefit to this whole paper. In fact, its message is solely, a wide variety of inputs is used to produce a wide variety of outputs.
- Fig. 3 I would guess this categorization can at most be understood in a fuzzy way. For example while FourCastNet does include a transformer backbone, its most important architectural design choice is the fourier neural operator, which as a spectral method does not actually fit into the categorization.
- Table 1 introduces arbitrary boundaries across a range of categories. Yet the only distinguishing factor between regional and global NWP models is that regional models use a regional domain.
- 4.3 SST does not cause El-Nino. Rather, El-Nino is a climatic regime, which you can diagnose by observing aggregates of SST over a certain region.
- Generalization Opportunity (2): The claim is refutable, there are a range of studies now which showcase AI-models to not necessarily display less skill under extreme events, e.g. tropical cyclones.
- RAG Opportunity (1) needs better motivation from an NWP perspective. Due to the chaotic nature of the atmosphere two almost identical initial states can lead to vastly diverging trajectories.
- RAG Opportunity (2), it is unclear what exactly the LLM should do, i.e. how it will help with NWP.
- 5.3 The sentence " In the weather domain, weather prediction can be formulated
as weather generation conditioned on temporal and spatial similarities." needs a reference or explanation. "partial differential continuity equations (Broomé & Ridenour, 2014; Palmer, 2019), which describe the weather as a flux" does not make sense, the weather is decribed by a set of physical quanitites, there time evolution is described with PDEs. Fluxes represent the change exchange of quantities between grid cells.
- 5.3 " Tobler’s First law of Geography (Tobler, 2004), which states that everything is related to everything else, but near things are more related than distant things;" is a proposition from geography, which just describes the existence of spatial autocorrelation of many key geographic phenomena. Prescribing certain spatial autocorrelation for NWP seems not a particularly fruitful idea, as the spatial autocorrelation can change rapidly over scales, locations and time.
- 5.3 "Tobler’s Second law of Geography (Tobler, 1999), which states that the phenomenon external to a geographic area of interest affects what goes on inside;" is a proposation from geography which just describes that the Earth system need be understood as a whole. Current global AI-based NWP models clearly embody this already, as they forecast the global field, including long-range interactions.
- 5.4 lacks references, there is a large body of literature on this
- 5.4 i don't understand how a knowledge graph obtained with a LLM from weather news reports should help in any way with NWP

---

> ### Author Response · Authors · 2025-04-27
> **Author Response (Part 1/5)**
>
> Thank you for the informative reviews. We provide rebuttals with **three** sections: "weakness", "comments and examples", "request changes". All changes are marked in blue in the revised manuscript.
>
> ### Weaknesses:
>
> > **W1:** The literature review is not systematic and misses out on relevant publications
>
> Added AtmoRep, as suggested. We have also included other relevant publications.
>
>
> > **W2:** The categorization of papers into bins of methodologies is at best loosely applicable, and more importantly, it is not clear how this categorization draws new connections or is otherwise particularly useful
>
> Our taxonomy is based on **training paradigms (deterministic predictive, probabilistic generative, and pretraining/fine-tuning)**, rather than ML architectures [1], diverse meteorological applications [2], or data modality [3], which have been reviewed elsewhere recently. Training paradigms often dictate the use of model architectures. In Figure 3, our intent was also to highlight emerging trends in model development, particularly the growing interest in probabilistic generative models and foundation models. While they currently represent a smaller fraction compared to the deterministic predictive models, their recent emergence underscores a new research focus.
>
>
> > **W3:** Large parts of the paper are short 1-sentence summaries of individual papers with little to no narrative behind.
>
> We have tried to address this concern as best as possible (see Section 4).
>
>
> > **W4:** The presentation and use of terms and ideas from meteorology and Earth system science ranges from poor to outright wrong.
>
> Thanks for pointing it out. We have improved the manuscript in consultation with domain experts from Climate and Earth sciences to ensure that all meteorological terms/concepts are described with technical accuracy. Please see highlighted changes in the manuscript.
>
>
> > **W5:** The presented future research directions are either extremely general and apply to almost all applications of machine learning (robustness, generalization, explainability, uncertainty quantification and data quality), suggest using a new development from mainstream ML also for weather forecasting (RAG, Multi-modal learning), have already been published quite a lot on weather forecasting (data storage/compresison, varying resolution) or are not properly explained and seem to misinterpret findings from meteorology and geography (GenAI with weather constraints).
>
> Given our focus on **training paradigms**, we believe that the future directions suggested are quite relevant. While the above issues or methods have been extensively studied in deterministic predictive models, they remain underexplored in the latter two paradigms, which are more recent. We summarize these possibilities as future directions. **Retrieval-Augmented Generation (RAG)**, originally developed in the natural language processing (NLP) domain, enables models to incorporate contextually relevant external knowledge. To date, we are not aware of its direct application in the weather forecasting domain. However, recent efforts in general time series forecasting [4–7] inspired us to highlight RAG as a promising direction in this survey. **Multi-modal learning** could involve image, text, audio, and time series data. For weather forecasting, this could integrate spatio-temporal data with textual description and prompt engineering techniques [7–8]. We have shown examples in the following responses. We have **removed** the section on GenAI with weather constraints, as suggested.
>
> *[1] Ren et al., Deep learning-based weather prediction: a survey. Big Data Research, 2021. \
> [2] Materia et al., Artificial intelligence for climate prediction of extremes: State of the art, challenges, and future perspectives. Wiley Interdisciplinary Reviews: Climate Change, 2024. \
> [3] Chen et al., Foundation models for weather and climate data understanding: A comprehensive survey. arXiv 2023. \
> [4] Liu et al., Retrieval-augmented diffusion models for time series forecasting. NeurIPS 2024. \
> [5] Ning et al., TS-RAG: Retrieval-Augmented Generation based Time Series Foundation Models are Stronger Zero-Shot Forecaster. arXiv 2025. \
> [6] Zhang et al., TimeRAF: Retrieval-Augmented Foundation Model for Zero-shot Time Series Forecasting. arXiv 2024. \
> [7] Yang et al., TimeRAG: BOOSTING LLM Time Series Forecasting via Retrieval-Augmented Generation. ICASSP 2025. \
> [8] Li et al., CLLMate: A Multimodal LLM for Weather and Climate Events Forecasting. arXiv 2024.*

---

> ### Author Response · Authors · 2025-04-27
> **Author Response (Part 2/5)**
>
> ### Comments and Examples (CE)
> > **CE1:** An example on the overly excessive use of verbosity in this text (**Abstracts**) \
> > **CE2:** the use of  real-world applications  in the abstract makes no sense (**Abstracts**) \
> > **CE3:** Motivating this survey with the issue of climate change seems not straight-forward, NWP is important regardless of climate change, I'd rather suggest to motivate why AI seems a good idea for NWP. (**Introduction**) \
> > **CE4:** The first two points on why conventional numerical solvers are hampered for NWP need more elaborate explanation (**Introduction**) \
> > **CE5:** Equation 1 brings no benefit to this whole paper. \
> > **CE6:** 5.4 lacks references, there is a large body of literature on this (**Section 5.4**) \
> > **CE7:** 4.3 SST does not cause El-Nino. Rather, El-Nino is a climatic regime, which you can diagnose by observing aggregates of SST over a certain region.  (**Section 4.3**)
>
> Thanks for the above comments. We have either rewritten or deleted text as appropriate. See changes in the introduction.
>
> > **CE8:** The third point is wrong: Current operational NWP systems are run in large ensembles to produce probabilistic weather forecasts. In fact, it is rather that most ML-based approaches (at least until ~2 years ago) have fallen short here, by just focussing on deterministic forecasting. (**Introduction**)
>
> We meant to say that a **single deterministic physics-based** model does not produce probabilistic outputs when initial conditions are fixed. We agree that ensemble forecasts can produce probabilistic weather forecasts, independent of whether they are physics-based or ML-based. We have rephrased that sentence to be more precise. Moreover, we have cited Gencast, a recent ML-based model (Nature 2025 [9]), which produces probabilistic forecasts by leveraging probabilistic diffusion techniques.
>
> *[9] Price et al., Probabilistic weather forecasting with machine learning. Nature, 2025.*
>
> > **CE9:** Introducing ARIMA immediately after conventional NWP systems seems unlogical.
>
> Given that ARIMA is a statistical model and falls outside the primary scope of this deep learning-focused survey, and considering its relative simplicity in application, we decided to remove it.
>
> > **CE10:** The Introduction discusses the advancements in AI-based NWP a bit to enthusiastically. For example, while PanguWeather and GraphCast are surely important contributions, it is less easy to say the outperform e.g. IFS already, as they did not produce probabilistic forecasts. Also the comparison in compute costs needs to be more nuanced, most AI-based NWP runs on GPUs while most conventional solvers run on CPUs.
>
> **Accuracy:** We only meant that PanguWeather and GraphCast outperform IFS in terms of accuracy on benchmark datasets, e.g., ERA5. We have rewritten the sentence for better clarity. Please refer to the responses to **CE8** for probabilistic forecasts.
>
> **Computational time:** We acknowledge that it is not easy to compare these models fairly since they primarily run on different architectures. However, the following table from [10], attempts a fairer comparison between ML models and conventional physics-based NWP models, e.g., IFS. We have accordingly revised the sentences.
>
> |                         |IFS HRES            |Pangu-Weather                | GraphCast               |
> |------------------|--------------------|-----------------------------| ------------------------|
> |Inference time  |∼ 50 minutes        |several seconds; single GPU  | ∼1 minute; single TPU   |
>
> *[10] Rasp et al., WeatherBench 2: A benchmark for the next generation of data-driven global weather models. arXiv 2024.*

---

> ### Author Response · Authors · 2025-04-27
> **Author Response (Part 3/5)**
>
> > **CE11:** Section 2.1 needs to explain clearly what reanalysis data is, and what original data sources are used to produce a reanalysis. Also the section totally misses out on radar, satellite, balloon and aircraft data, which are arguably at least as important as station data.
>
> We have provided a clearer explanation of the observed data, reanalysis data, and original data sources. We have also added radar and satellite data. See highlighted changes (in blue) in Section 3.1 in the revised manuscript.
>
> > **CE12:**   Section 2.2 leaves the reader with a false impression of NWP. Only very rarely are forecasts made specifically for particular extreme events. Rather NWP forecasts are produced, and afterwards analyzed for the prevalence of extremes.
>
> Section 2.2 (Section 3.2 in the current version) discusses weather forecasting tasks from **various perspectives**: temporal, spatial, applications to meteorological variables, and event types (extreme or normal cases). They do not refer to any specific NWP models. Additionally, a similar discussion can be also found from the [ClimaX](https://proceedings.mlr.press/v202/nguyen23a/nguyen23a.pdf) paper, which was accepted by ICML 2023.
>
> >  **CE13:**  Fig. 3 I would guess this categorization can at most be understood in a fuzzy way. For example while FourCastNet does include a transformer backbone, its most important architectural design choice is the fourier neural operator, which as a spectral method does not actually fit into the categorization.
>
> Agree some models do not fit into the fine-grained categorization, especially when they employ multiple models/techniques. However, the main categorization of this survey focuses on **training paradigm**. FourCastNet employs the Transformer architecture for **deterministic** weather forecasting over multiple variables across the globe, which is why it is put into *Deterministic predictive learning --> General-Purpose Large Models --> Transformer*.
>
> **Note:** Figure 1 in the current version of the manuscript was moved to a prior section to give the readers an earlier preview into the taxonomy, as suggested by reviewer 1bNc.
>
> >   **CE14:**  Table 1 introduces arbitrary boundaries across a range of categories. Yet the only distinguishing factor between regional and global NWP models is that regional models use a regional domain.
>
> While the study scope is clearly different for global and regional models, they employ different **types of training data** in terms of the spatial/temporal resolution and other features. Global forecasting often focuses on general trends using low-resolution data, while regional forecasting tends to use localized and fine-grained data. Our survey highlights these distinctions in spatial and temporal resolutions, training data duration, and DL architectures. Please see details in the paragraphs just before Table 1 in the revised manuscript.
>
> > **CE15:** Generalization Opportunity (2): The claim is refutable, there are a range of studies now which showcase AI-models to not necessarily display less skill under extreme events, e.g. tropical cyclones.
>
> We highlight the comparison between extreme and non-extreme events. Compared to **non-extreme** events, ML models still perform **poorly in forecasting extreme events** due to their rarity and limited representation in the training data [11-12], even though efforts are ongoing.
>
> *[11] Camps-Valls et al., Artificial intelligence for modeling and understanding extreme weather and climate events. Nature Communications, 2025.\
> [12] Materia et al., Artificial intelligence for climate prediction of extremes: State of the art, challenges, and future perspectives. Wiley Interdisciplinary Reviews: Climate Change, 2024.*

---

> ### Author Response · Authors · 2025-04-27
> **Author Response (Part 4/5)**
>
> > **CE16:**  RAG Opportunity (1) needs better motivation from an NWP perspective. Due to the chaotic nature of the atmosphere two almost identical initial states can lead to vastly diverging trajectories.
>
> The concern makes sense. However, we contend that given enough “context”, identical initial states are more likely to lead to converging trajectories. The context may need to be both spatial and temporal. Our intention in highlighting the opportunity of RAG is to enhance the model performance using a **data-centric perspective**.
>
> Specifically, we see two key advantages of RAG for weather forecasting or even more general time series forecasting: (1) **Explicit access to relevant historical patterns** at inference time: retrieved information can directly guide prediction, making relevant past cases available without relying solely on patterns embedded in model weights. (2) **Compensating for the rarity of or extreme events**: poor performance on extreme events can be traced to their underrepresentation in training data. A retrieval module allows the model to dynamically access similar historical instances when they reoccur, thereby improving generalization and robustness in forecasting extreme events.
>
> We have clarified this in the revised manuscript (see Section 6.2) to better align our motivation with the forecasting context.
>
> > **CE17:** RAG Opportunity (2), it is unclear what exactly the LLM should do, i.e. how it will help with NWP.
>
> LLMs could do two things: (1) **Impact Reasoning**: By grounding forecasts in historical event narratives (news articles) and consequences (e.g., floods, infrastructure damage), LLMs can reason about **contextualized, downstream impacts** that NWP alone cannot capture. (2) **Semantic Mapping**: LLMs learn to map complex meteorological patterns (e.g., from ERA5 raster grids) to natural language event categories, effectively automating what is traditionally the role of expert meteorologists interpreting raw outputs. LLMs would work together with tradiction NWP models rather than replace them. A recent related work is [*CLLMate: A Multimodal Benchmark for Weather and Climate Events Forecasting*](https://arxiv.org/abs/2409.19058), which leverages both text and numerical data.
>
> > **CE18:** 5.3 The sentence " In the weather domain, weather prediction can be formulated as weather generation conditioned on temporal and spatial similarities." needs a reference or explanation. \
> > **CE19:** 5.3  "Tobler’s First law of Geography (Tobler, 2004). Prescribing certain spatial autocorrelation for NWP seems not a particularly fruitful idea, as the spatial autocorrelation can change rapidly over scales, locations and time. \
> > **CE20:** 5.3 "Tobler’s Second law of Geography (Tobler, 1999). Current global AI-based NWP models clearly embody this already, as they forecast the global field, including long-range interactions.
>
> We appreciate the constructive comments. We decided to remove Section 5.3, as suggested in the requested change.
>
> > **CE21:** 5.4 i don't understand how a knowledge graph obtained with a LLM from weather news reports should help in any way with NWP
>
> CLLMate [13], a recent publication, aligns numerical meteorological raster data with textual event data and leverages LLMs to predict weather and climate events, demonstrating how these two data modalities can effectively complement each other. Additionally, [14] develops a knowledge graph framework to automatically generate an overview of weather events, including the prediction and reasoning tasks. Motivated by these advances, we believe that combining knowledge graph and numerical weather data is an interesting research direction. We have revised Section 6.3 accordingly.
>
> *[13] Li et al., CLLMate: A Multimodal Benchmark for Weather and Climate Events Forecasting. arXiv, 2025. \
> [14] Qu et al., Knowledge Graph-Driven Weather Overview Generation for the Beijing 2022 Winter Olympic Games. Journal of Meteorological Research, 2024.*

---

> ### Author Response · Authors · 2025-04-27
> **Author Response (Part 5/5)**
>
> ### Requested Changes (RC)
>
> > **RC1:** Introduce a proper systematic methodology for the literature review, e.g. by querying existing literature databases with keyword search or document embeddings.
>
> We, somehow, used the method you mentioned here and categorized those models in Figure 1. Meanwhile, we added more analysis of different models in Section 4. Only ad hoc methods were used for the literature search. However, all references cited in the publications were scanned for relevance to this survey.
>
> > **RC2:** Remove the contributions in the introduction
>
> Rewrote the summary paragraph at the end of the introduction.
>
>
> > **RC3:** Add missing papers (e.g. AtmoRep) \
> >  **RC4:** Properly explain terms related to NWP \
> >  **RC5:** Properly explain how the mentioned applications relate to NWP \
> > **RC6:** Develop a key message for this survey \
> > **RC7:** Rewrite the abstract and introduction to be less verbose, but rather clearly display the key message of this survey
>
> The above requested changes were accepted. Please see the changes in the manuscript.
>
> > **RC8:** Restructure the categorization and rewrite the article, such that new connections between the surveyed articles become evident
>
> We have tried to address this concern as best as possible.
>
>
> > **RC9:** Remove generic points from the presented future directions \
> > **RC10:** Introduce new future directions, which are not yet well-studied in current literature, but rather arise from problems of current approaches - from conventional NWP and also from AI-based NWP
>
> Please refer to the responses to **W5** above. Additionally, we have included Section 6.5 about **model compression with the distillation technique** for better efficiency.
>
> > **RC11:** Remove the points related to GenAI with weather constraints
>
> Removed.
>
> > **RC12:** Remove the section in the introduction that focusses on other surveys
>
> Moved this to “Section 2: Related Surveys”
>
> >  **RC13:** Add proper explanation of the different approaches of conventional NWP. Especially make clear the differences and commonalities between (radar) nowcasting, medium-range weather forecasting and seasonal prediction.
>
> Added explanations in Section 3.2.
>
> > **RC14:** Please note: for most weather prediction tasks, it is enough to look at benchmarks to understand which methods are SOTA. Hence, this alone is not a meaningful contribution of a survey.
>
> The last paragraph in the introduction has been rewritten.

---

### Decision · Action_Editor_mZsH · 2025-06-12

**Recommendation:** Reject

**Audience:**

Yes

**Audience Explanation:**

While the topic may be of interest to a small subset of TMLR's audience, we feel that the current submission may not be the best fit for the journal. Although TMLR does consider survey papers, it prioritizes those that draw novel connections, highlight emerging trends, or identify new research directions. Additionally, the short article format limits the depth and nuance expected for such contributions.

**Claims And Evidence:**

No

**Claims Explanation:**

After some discussion, the reviewer and action editor agreed that the claims made in the submission are not sufficiently supported by accurate, convincing, and clear evidence, concretely:

- While the effort to categorize existing work is appreciated, the proposed taxonomy appears loosely defined and inconsistently applied. Several examples, such as Conformer (Saleem et al., 2024) and AtmoRep (Lessig et al., 2023), illustrate that papers often do not fit neatly into the suggested categories. Moreover, the boundaries between categories—e.g., between model architectures or learning paradigms—are not clearly delineated, making the taxonomy difficult to apply consistently. In practice, many models combine ideas from multiple approaches, further highlighting the non-mutually exclusive nature of the classification.

- Similarly, the discussion on future research directions, while broadly relevant, remains too general to be actionable. Suggestions such as using hybrid ML models to improve interpretability lack concrete justification or supporting evidence. Without a clearer rationale or specific examples, these claims do not provide sufficient guidance for advancing the field.

Overall, the manuscript would benefit from a more precise and evidence-based articulation of both its taxonomy and its proposed future directions.